# SongEcho: Towards Cover Song Generation via Instance-Adaptive Element-wise Linear Modulation

**Sifei Li[1,2], Yang Li[1,2], Zizhou Wang[2], Yuxin Zhang[1,2], Fuzhang Wu[3], Oliver Deussen[4], Tong-Yee Lee[5], Weiming Dong[1,2]** [*]

[1]MAIS, Institute of Automation, Chinese Academy of Sciences
[2]School of Artificial Intelligence, University of Chinese Academy of Sciences
[3]ISRC, Institute of Software, Chinese Academy of Sciences
[4]University of Konstanz
[5]National Cheng-Kung University
{lisifei2022, yangli2022, weiming.dong}@ia.ac.cn

## Abstract

Cover songs constitute a vital aspect of musical culture, preserving the core melody of an original composition while reinterpreting it to infuse novel emotional depth and thematic emphasis. Although prior research has explored the reinterpretation of instrumental music through melody-conditioned text-to-music models, the task of cover song generation remains largely unaddressed. In this work, we reformulate our cover song generation as a conditional generation, which simultaneously generates new vocals and accompaniment conditioned on the original vocal melody and text prompts. To this end, we present **SongEcho**, which leverages **I**nstance-**A**daptive **E**lement-wise **L**inear **M**odulation (**IA-EiLM**), a framework that incorporates controllable generation by improving both conditioning injection mechanism and conditional representation. To enhance the conditioning injection mechanism, we extend Feature-wise Linear Modulation (FiLM) to an **E**lement-wise **L**inear **M**odulation (**EiLM**), to facilitate precise temporal alignment in melody control. For conditional representations, we propose **I**nstance-**A**daptive **C**ondition **R**efinement (**IACR**), which refines conditioning features by interacting with the hidden states of the generative model, yielding instance-adaptive conditioning. Additionally, to address the scarcity of large-scale, open-source full-song datasets, we construct **Suno70k**, a high-quality AI song dataset enriched with comprehensive annotations. Experimental results across multiple datasets demonstrate that our approach generates superior cover songs compared to existing methods, while requiring fewer than 30% of the trainable parameters. The code, dataset, and demos are available at https://github.com/lsfhuihuiff/SongEcho_ICLR2026.

## 1 Introduction

If great melodies merit reinterpretation, then exceptional cover songs breathe new life into their originals. Cover songs play an essential role in musical culture, acting as conduits for cultural memory and agents in the formation of a musical canon. Iconic examples, such as Whitney Houston's transformative rendition of Dolly Parton's "I Will Always Love You", reinterpret the style of the song, evolving a gentle country ballad into a worldwide anthem of deep affection [1]. Given the expressive potential of musical reimagination and cultural significance, we think that cover song generation is a field worthy of exploration.

Similar to Whitney Houston's rendition, musicians creating cover songs may introduce flexible adaptations in local musical elements, such as phoneme durations, vibrato, and note transitions,

---

[*]Corresponding author.
[1]https://www.youtube.com/shorts/PdE_dAkMDW4

yielding highly varied and personalized reinterpretations. In this work, we abstract a cover paradigm applicable to arbitrary songs and reformulate our cover song generation as a conditional generation task that performs a global style transfer with text guidance while preserving the source vocal melody contour and excluding local customized adaptations. Specifically, the task requires a model to leverage the provided vocal melody as a foundation structure, while concurrently synthesizing vocal and harmonious accompaniment that aligns with a given text prompt. Although text-to-song generation has advanced considerably, the task of cover song generation remains largely unaddressed. The primary challenge lies in devising a model that can implicitly disentangle vocal components, ensure temporal melody control and lyric synchronization, and produce coherent accompaniment. Neglecting these elements may result in misaligned lyrics, inconsistent melodies, or degraded audio quality, thereby necessitating robust capabilities in collaborative generation and content control.

This task differs from Singing Voice Synthesis (Liu et al., 2022; Cui et al., 2024; Zhang et al., 2024a) and Singing Voice Conversion (Ferreira et al., 2025; Jayashankar et al., 2023), which deal with single-track vocals and focus on short audio segments (5-20s) that can be concatenated to produce longer audio. In contrast, cover song generation simultaneously synthesizes vocals and accompaniment, necessitating coherence of the accompaniment across the entire song.

Recent works (Wu et al., 2024; Ciranni et al., 2025; Tsai et al., 2025) have achieved melody control in pretrained text-to-music models, demonstrating potential applicability to cover song generation. The core difference among these methods lies in their melody condition injection mechanisms, employing either cross-attention (Tsai et al., 2025) or element-wise addition (Ciranni et al., 2025; Wu et al., 2024) (see Figures 1(a) and 1(b)). Cross-attention mechanisms require extra modeling of temporal alignments, which is inherently indirect and introduces computational redundancy across potentially misaligned dimensions. Element-wise addition leverages the temporal correspondence between sequences but limits modulation flexibility, acting as an affine transformation with a fixed scaling factor. Beyond these limitations in condition injection mechanisms, existing methods independently encode melody conditions, thereby failing to provide targeted adaptation to the generative model's hidden states. Consequently, incompatible condition vectors may distort the hidden states during condition injection, resulting in unnatural and low-fidelity audio synthesis.

To address the aforementioned challenges, we present a novel framework, SongEcho, for cover song generation built upon a text-to-song model (Gong et al., 2025). We propose Instance-Adaptive Element-wise Linear Modulation (IA-EiLM), which comprises Element-wise Linear Modulation (EiLM) and Instance-Adaptive Condition Refinement (IACR). These components enhance controllable generation by refining the condition injection mechanism and conditional representation. (1) Injection Mechanism: Feature-wise Linear Modulation (FiLM) (Perez et al., 2018) has demonstrated efficacy as a conditioning technique. Birnbaum et al. (2019) proposed TFiLM, which temporally applies FiLM by partitioning sequences into blocks and using an RNN (Elman, 1990; Graves, 2012) to recurrently generate block-wise modulation parameters. In contrast, we extend FiLM to EiLM (see Figure 1(c)), which generates modulation parameters matching the target dimensions in a single operation without temporal dependency. This design enables element-wise modulation of hidden states, ensuring the temporally aligned injection of melody. (2) Conditional Representation: We introduce the IACR module to rectify the rigidity of traditional condition encoding. By enabling interaction between hidden states and external conditions, IACR dynamically adapts conditions to the hidden states, mitigating feature conflicts and audio quality degradation caused by static condition injection.

Our contributions can be summarized as follows:

- We introduce SongEcho, a parameter-efficient framework that enables cover song generation by leveraging a novel conditioning method that achieves fine-grained control of the vocal melody.

- We propose Instance-Adaptive Element-wise Linear Modulation (IA-EiLM), which comprises the EiLM and Instance-Adaptive Condition Refinement (IACR), enhancing the condition injection mechanism and conditional representation, respectively.

- To address the lack of open-source, high-quality, large-scale full-song datasets, we introduce Suno70k, an open-source AI song dataset enriched with detailed annotations, including enhanced tags and lyrics.

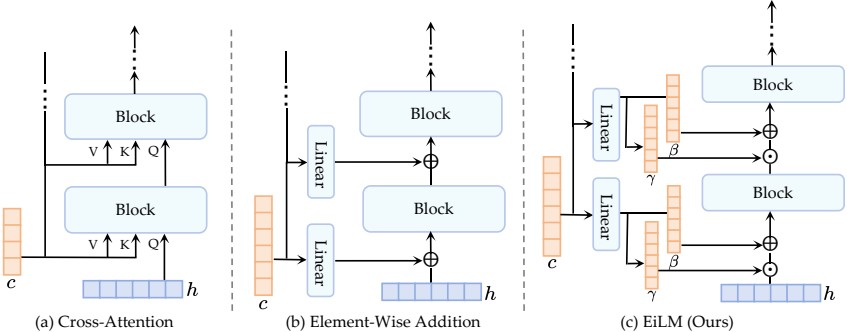

Figure 1: Differences between other condition injection mechanisms and our approach. EiLM eliminates the need for separate learning of temporal alignment in (a) while offering more flexible modulation than (b). "$\oplus$" represents element-wise addition, and "$\odot$" represents element-wise multiplication.

- Experimental results demonstrate that our method generates superior cover songs, outperforming state-of-the-art approaches across all metrics on multiple datasets.

## 2 RELATED WORK

**Text-to-Song Generation.** Jukebox (Dhariwal et al., 2020) pioneered song generation. In recent years, industry tools such as Suno[2], Udio[3], Seed-Music (Bai et al., 2024), and Meruka[4] have shown promising results in this domain. Academic efforts have followed closely, with language model-based song generation approaches, including Melodist (Hong et al., 2024), Melody (Li et al., 2024a), Songcreator (Lei et al., 2024), YuE (Yuan et al., 2025), SongGen (Liu et al.), and LeVo (Lei et al., 2025), which autoregressively generate song tokens but require significant inference time. Diffusion-based methods, such as DiffRhythm (Ning et al., 2025) and ACE-Step (Gong et al., 2025), have substantially reduced this latency. Notably, ACE-Step (Gong et al., 2025) improves upon DiffRhythm (Ning et al., 2025) by incorporating song structure understanding. Although current models generate high-quality songs and some support audio prompts (Yuan et al., 2025; Lei et al., 2025), they lack the capability for precise temporal melody control. Considering both inference speed and performance, we adopt ACE-Step as our base model.

**Singing Voice Synthesis & Conversion.** Extensive research in Singing Voice Synthesis (SVS) (Zhang et al., 2023b; Liu et al., 2022; Zhang et al., 2024a; 2025) and Singing Voice Conversion (SVC) (Lu et al., 2024; Chen et al., 2024; Ferreira et al., 2025; Shao et al., 2025) has led to significant progress in generating high-quality, controllable single-track vocals. Nonetheless, these approaches are inherently limited as they do not address the generation of instrumental accompaniment. This work, in contrast, tackles the more holistic problem of full-song generation, requiring the simultaneous synthesis of a vocal track and its coherent accompaniment.

**Controllable Music Generation.** Recent work has advanced controllable music generation by incorporating temporal conditions into text-to-music models using various approaches, such as In-attention (Lan et al., 2024), ControlNet-style addition (Wu et al., 2024; Ciranni et al., 2025; Hou et al., 2025), and cross-attention (Tsai et al., 2025; Lin et al., 2024; Yang et al., 2025). However, these dominant paradigms exhibit significant trade-offs: additive methods offer limited modulation flexibility, while cross-attention is indirect and computationally redundant. Critically, all these approaches encode the condition in isolation, lacking a mechanism to dynamically adapt the conditional signal to the generator's internal hidden states. In contrast, our approach addresses the aforementioned issues by improving the condition injection mechanism and enhancing conditional representations.

---

[2] https://suno.com/blog/introducing-v4-5
[3] https://www.udio.com/blog/introducing-v1-5
[4] https://www.mureka.ai

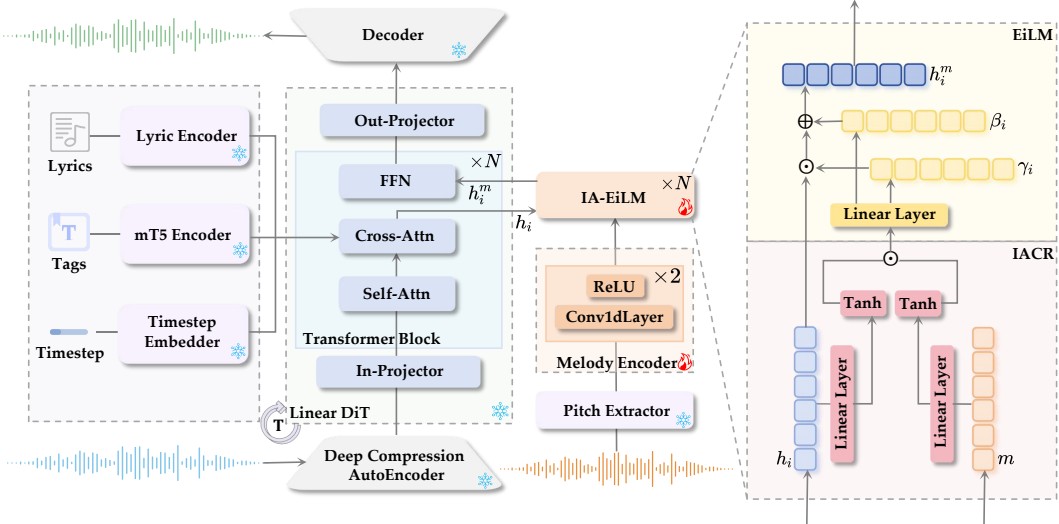

Figure 2: We employ a Diffusion Transformer (DiT) as the song generation backbone with a novel conditioning method, "IA-EiLM", for vocal melody control. A Pitch Extractor and Melody Encoder extract melody features, denoted as "$m$". The IA-EiLM module, integrated into each Transformer block, comprises two components: IACR and EiLM. "IACR" facilitates interaction between "$m$" and hidden states "$h_i$", refining melody condition, while "EiLM" modulates "$h_i$" into "$h_i^m$" with modulation parameters "$\gamma_i$" and "$\beta_i$", derived from the refined melody condition.

**Conditional Normalization.** Conditional Normalization methods are a powerful class of techniques that inject information by learning the parameters for an affine transformation of a network's intermediate features. Unified by frameworks like FiLM (Perez et al., 2018), this approach has been highly successful in a wide range of domains, including image style transfer (Dumoulin et al., 2017; Huang & Belongie, 2017), semantic image synthesis (Park et al., 2019), speech recognition (Kim et al., 2017), modern text-to-image models (Peebles & Xie, 2023), text classification (Birnbaum et al., 2019), and black-box audio effect modelling Comunità et al. (2023). However, its application to music remains unexplored, and our work investigates its potential in this domain.

## 3 METHOD

We propose SongEcho, a parameter-efficient framework for our cover song generation, built upon the full-song generation model ACE-Step (Gong et al., 2025) and leveraging the Instance-Adaptive Element-wise Linear Modulation (IA-EiLM), as illustrated in Figure 2. We start by introducing IA-EiLM and then describe our particular model for cover song generation.

### 3.1 ELEMENT-WISE LINEAR MODULATION (EILM)

Unlike prior temporally controllable music generation methods (Wu et al., 2024; Tsai et al., 2025; Hou et al., 2025; Yang et al., 2025), which rely on cross-attention or element-wise addition, we explore the application of FiLM (Perez et al., 2018) for melody injection and extend it to EiLM.

Let $c \in \mathbb{R}^{B \times T \times M}$ denote a condition feature, where $B$ is the batch size, $T$ is the sequence length, and $M$ is the condition dimension. Let $h_i \in \mathbb{R}^{B \times T \times D_i}$ represent the hidden states of the $i$-th layer of the generative backbone, where $D_i$ is the number of feature dimensions in the layer. We aim to learn a mapping function that modulates $h_i$ using $c$ to generate a cover song. Feature-wise Linear Modulation (FiLM) is an effective conditioning method that has not yet been applied to controllable music generation. To enable precise temporal control, we propose Element-wise Linear Modulation (EiLM) as an extension of Feature-wise Linear Modulation (FiLM). This conditional modulation method dynamically adapts hidden states to melody conditions through a time-varying

affine transformation. The overall modulation is defined as:

$$h_i^m = \text{EiLM}(h_i|c) = \gamma_i \odot h_i + \beta_i, \tag{1}$$

$$(\gamma_i, \beta_i) = f_i(c), \tag{2}$$

where $\gamma_i, \beta_i \in \mathbb{R}^{B \times T \times D_i}$ are the modulation parameters, derived from $c$ via a linear projector $f_i$. $h_i^m \in \mathbb{R}^{B \times T \times D_i}$ is the modulated hidden states. Our EiLM generalizes FiLM (Perez et al., 2018) by generating modulation parameters that precisely match the shape of the hidden states.

## 3.2 INSTANCE-ADAPTIVE CONDITION REFINEMENT (IACR)

In addition to external improvements to the condition injection mechanism, we propose a condition refinement strategy, termed Instance-Adaptive Condition Refinement (IACR), which adaptively refines the condition vector based on the hidden states of the generative backbone for improving conditional representations. Our IACR module employs a gating mechanism adapted from WaveNet (van den Oord et al., 2016), where we enable cross-modal interaction between two branches. Beyond merely encoding the conditional input, our method ensures that the conditional features dynamically adapt to the hidden states. Specifically, a vocal pitch sequence $p \in \mathbb{R}^{B \times T^0 \times 1}$ is first processed by a melody encoder to produce melody features $m \in \mathbb{R}^{B \times T \times M}$. $m$ are then interactively refined with the hidden states $h_i \in \mathbb{R}^{B \times T \times D_i}$ via a gating mechanism (Van den Oord et al., 2016), denoted as:

$$h_i' = L_{h_i}(h_i), \quad m_i' = L_{m_i}(m) \tag{3}$$

$$c_i = \tanh(h_i') \odot \tanh(m_i'), \tag{4}$$

where $L_{h_i}, L_{m_i}$ denote linear layers, $h_i' \in \mathbb{R}^{B \times T \times M}$, and $m_i' \in \mathbb{R}^{B \times T \times M}$. The refined condition $c_i \in \mathbb{R}^{B \times T \times M}$ dynamically adapts to the current generative instance, enabling selective integration and interpretation of the melodic features.

**Why is IACR necessary?** To the best of our knowledge, existing control injection methods derive conditional features solely from the conditional input, overlooking their compatibility with the generative model's hidden state. We take our EiLM as an example to demonstrate the necessity of IACR.

In text-to-song models, the hidden states are not a blank canvas. Formally, a hidden state $h = \epsilon_\theta(t_{tag}, l, t) \in \mathbb{R}^{B \times T \times D}$, conditioned on a text prompt $t_{tag}$, lyrics $l$, and timestep $t$, already embeds an intrinsic melodic structure $M_h$. The goal of conditioning is to modulate $h$ with parameters $(\gamma, \beta)$ such that the melody of the output, $M_c \approx E_m(\gamma \odot h + \beta)$, where $E_m$ is a hypothetical melody encoder that extracts melody from the hidden states and the target melody $M_c$ is derived from a melody feature $m$.

A conventional static conditioning approach generates modulation parameters solely from the melody feature $m$ as follows:

$$(\gamma_m, \beta_m) = F(m), \tag{5}$$

where $F$ denotes the conditional mapping function. The optimization problem can be formulated as:

$$(\gamma_m, \beta_m) = \arg\min_{\gamma, \beta} \|E_m(\gamma \odot h + \beta) - M_c\|_2^2. \tag{6}$$

Given $\gamma_m \neq 0$ in our task ($\gamma_m = 0$ loses timbre and lyrics), without access to the hidden states $h$ or their intrinsic melody $M_h$, the transformation network $T$ must learn a universal mapping $\Delta_{M_h \to M_c}$ across all possible $h$, causing Equation 6 to be underconstrained.

In contrast, our instance-adaptive conditioning approach, implemented in the IACR module, computes the parameters based on both $m$ and $h$ as:

$$(\gamma_{h,m}, \beta_{h,m}) = F(m, h). \tag{7}$$

By providing the network $F$ with direct access to $h$, the task is transformed into a one-to-one mapping problem. In this context, $\gamma_{h,m}$ and $\beta_{h,m}$, encoding both melodic conditions and hidden states, expand the conditional representation space. Tailored to hidden states, these conditions enable seamless integration into the generative model, thereby improving melodic control and audio quality (see in 5.4).

### 3.3 SONGECHO

Our proposed framework, **SongEcho**, extends the pre-trained text-to-song model, **ACE-Step**, by incorporating a melody encoder, $\mathcal{E}$, and integrating an **IA-EiLM** module into each transformer block. Given a vocal pitch sequence $p \in \mathbb{R}^{B \times T^0 \times 1}$, extracted at 100 Hz via RVMPE (Wei et al., 2023), $\mathcal{E}$, comprising 1D convolutional layers, encodes features as:

$$m^0 = \mathcal{E}(p), \quad m^0 \in \mathbb{R}^{B \times T^0 \times M}. \tag{8}$$

These are interpolated to align with the hidden states $h_i \in \mathbb{R}^{B \times T \times D_i}$, given by:

$$m = \text{Interpolate}(m^0), \quad m \in \mathbb{R}^{B \times T \times M}. \tag{9}$$

The features $m$ are then refined via State-Adaptive Condition Refinement (IACR) as follows:

$$c_i = \text{IACR}(m, h_i), \quad m \in \mathbb{R}^{B \times T \times M}. \tag{10}$$

Similar to Zhang et al. (2023a), to mitigate noise modulation in the hidden states caused by randomly initialized parameters, we initialize $f_i$ with zeros to ensure training starts from the original model. To incorporate zero initialization, we reformulate EiLM as follows:

$$\text{EiLM-zero}(h_i|c_i) = (\gamma_i + 1) \odot h_i + \beta_i, \tag{11}$$
$$(\gamma_i, \beta_i) = f_i(c_i). \tag{12}$$

Given that self-attention facilitates global information interaction across tokens, while the FFN layer performs localized feature transformations, we insert the IA-EiLM module before the FFN layer in each Transformer block to inject melody information and prevent its dilution within the global attention mechanism, as illustrated in Figure 2.

Except for $\mathcal{E}$ and the IA-EiLM modules, all model parameters are frozen. The training objective is defined as:

$$\mathcal{L}_{\text{FM}} = \mathbb{E}_{x_0, z \sim \mathcal{N}(0,I), t \sim U[0,1]} \left[ \|(\epsilon_\theta(x_t, t_{tag}, l, t, p) \cdot (-\sigma_t) + x_t) - x_0\|_2^2 \right], \tag{13}$$

where $x_0$ denote the latent representation, $x_t = (1 - \sigma_t)x_0 + \sigma_t z$. Since we do not update parameters related to semantic alignment, we disable the semantic alignment loss based on self-supervised learning models (Li et al., 2024c; Zanon Boito et al., 2024). Overall, our proposed method introduces vocal melody control in a lightweight manner. A detailed comparison is provided in Table 1.

## 4 DATASET

Due to copyright constraints, the availability of publicly accessible song datasets (Hsu & Jang, 2009; Bertin-Mahieux et al., 2011; Zhang et al., 2024b; Yao et al., 2025) remains significantly restricted (see details in the Appendix A.5). To address these limitations, we introduce Suno70k, a high-quality AI song dataset derived from the Suno.ai Music Generation dataset [5]. This open-source collection contains metadata, including song links, for 659,788 AI-generated songs, but the quality varies widely. Our curation process involves several steps:

**1. Data Filtering.** We filter the dataset based on metadata, removing entries with incomplete information (e.g., missing IDs, lyrics, or tags) and deduplicating by ID. We exclude purely instrumental tracks and entries with unclear lyric structures, unrecognizable characters, or non-English lyrics. To align with the 4-minute generation limit of the ACE-Step (Gong et al., 2025), we exclude all samples with a duration exceeding this threshold.

**2. Quality Assessment.** We download the corresponding audio files and evaluate them with SongEval (Yao et al., 2025) across five dimensions: overall coherence, memorability, naturalness of vocal breathing and phrasing, clarity of song structure, and overall musicality. Samples scoring below 3 (out of 5) in any dimension are excluded.

**3. Enhanced Tagging.** Observing that the tags from metadata are incomplete, we employ Qwen2-audio (Chu et al., 2024) to generate comprehensive tags across the following aspects: genre, vocal

---

[5] https://huggingface.co/datasets/nyuuzyou/suno

type, instruments, and mood. These are concatenated with the original tags, deduplicated, and limited to 20 tags per song, separated by commas, consistent with the official examples of ACE-Step (Gong et al., 2025).

In the end, we obtain a total of 69,469 songs, with 69,379 for training and 90 for testing, yielding a total duration of approximately 3,000 hours.

## 5 EXPERIMENTS

### 5.1 IMPLEMENTATION DETAILS

We employ our IA-EiLM on the open-source text-to-song model ACE-Step (Gong et al., 2025), a Linear Diffusion Transformer (DiT) (Xie et al., 2025) capable of generating high-quality songs efficiently. We freeze the parameters of the Linear DiT, lyric encoder, and text encoder, training only the IA-EiLM and Melody Encoder parameters. The learning rate is set to 1e-4 with a linear warm-up over 1,000 steps. We utilize the AdamW optimizer with $\beta_1 = 0.9$, $\beta_2 = 0.95$, and a weight decay of 0.01. The maximum duration for music generation is set to 240 seconds, consistent with ACE-Step. Experiments are conducted on three NVIDIA A100 GPUs for 30,000 steps with a batch size of 12 (1 per GPU with a gradient accumulation factor of 4).

### 5.2 EVALUATION METRICS

We develop a comprehensive evaluation protocol that includes the following metrics. For melody control, we extract melodies from ground-truth and generated songs and compute three metrics using the mir_eval library (Raffel et al., 2014): Raw Pitch Accuracy (RPA), the fraction of melody frames with pitch within half a semitone of the reference; Raw Chroma Accuracy (RCA), pitch accuracy ignoring octave; and Overall Accuracy (OA), the fraction of all frames correctly estimated, including pitch and voicing (melody vs. non-melody) alignment. Additionally, we adopt the open-source code[6] to calculate $FD_{openl3}$ (Cramer et al., 2019), $KL_{passt}$ (Koutini et al., 2022), and CLAP score (Wu et al., 2023). We use $FD_{openl3}$ and $KL_{passt}$ to assess differences between the generated music and the ground-truth distribution. The CLAP score evaluates consistency between the generated songs and their corresponding text tags. Furthermore, we compute the Phoneme Error Rate (PER) using Whisper (Radford et al., 2023) to evaluate the vocal content of the generated songs.

### 5.3 COMPARISON WITH STATE-OF-THE-ART METHODS

We compare our method with two state-of-the-art melody-guided music generation approaches: Stable Audio (SA) ControlNet (Hou et al., 2025) and MuseControlLite (Tsai et al., 2025). The former integrates ControlNet (Zhang et al., 2023a) into the DiT-based music generation model Stable Audio (Evans et al., 2024), while the latter employs the IP-adapter concept to enable melody control for Stable Audio. As both methods support only instrumental music generation, we apply them to the same base model, ACE-Step, used in our approach, and ensure consistency in the melody encoder. Since integrating ControlNet with ACE-Step requires over 80 GB of GPU memory at a batch size of 1, we adopt LoRA (Hu et al., 2022) fine-tuning for a subset of the copied branches, maximizing trainable parameters with a rank of 512. For reference, we also evaluate the performance of the original model. The number of trainable parameters for the three methods is shown in Table 1. Our method significantly reduces the trainable parameters, accounting for only 3.07% of ACE-Step+SA ControlNet, 14.8% of ACE-Step+SA ControlNet+LoRA and 26.0% of ACE-Step+MuseControlLite's parameters. Additional aesthetics evaluation of the comparison results is provided in Appendix C.3.

#### 5.3.1 QUANTITATIVE EVALUATION

**Objective Evaluation.** The performance of our method on the Suno70k test set is shown in Table 1. Our approach achieves superior results compared with the baselines. Notably, it demonstrates a clear advantage in Raw Pitch Accuracy (RPA) and Raw Chroma Accuracy (RCA). For the $FD_{openl3}$ metric, our method achieves reductions of 57.6% and 41.6% compared to ACE-Step+SA

---

[6] https://github.com/Stability-AI/stable-audio-metrics?spm=5d4b8e0. 56c16f66.0.0.310b73e8StYpFt

Table 1: Quantitative evaluation results on Suno70k test set. "TP" represents trainable parameters. The best results are in highlighted **bold** and the second best ones are underlined (same in the following tables).

| | RPA ↑ | RCA ↑ | OA ↑ | CLAP ↑ | FD ↓ | KL ↓ | PER ↓ | TP ↓ |
|---|---|---|---|---|---|---|---|---|
| ACE-Step (Gong et al., 2025) | - | - | - | 0.2930 | 73.53 | 0.2670 | 0.4168 | - |
| ACE-Step+SA ControlNet (Hou et al., 2025) | 0.6209 | 0.6440 | 0.6858 | 0.2875 | 105.95 | 0.2019 | 0.3714 | 1.6B |
| ACE-Step+SA ControlNet+LoRA (Hou et al., 2025) | 0.6214 | 0.6431 | 0.6833 | 0.2892 | 99.19 | 0.1850 | 0.3734 | 331M |
| ACE-Step+MuseControlLite (Tsai et al., 2025) | 0.5205 | 0.5346 | 0.5940 | 0.2977 | 72.04 | 0.2151 | 0.4194 | 189M |
| SongEcho (Ours) | **0.7080** | **0.7339** | **0.6952** | **0.3243** | **42.06** | **0.1123** | **0.2951** | **49.1M** |

ControlNet+LoRA and ACE-Step+MuseControlLite, respectively, highlighting its effectiveness in optimizing music quality while achieving melody control.

Table 2: Quantitative evaluation results on Suno70k test set with swapped tags.

| | RPA ↑ | RCA ↑ | OA ↑ | CLAP ↑ | FD ↓ | KL ↓ | PER ↓ |
|---|---|---|---|---|---|---|---|
| ACE-Step ( Gong et al. (2025)) | - | - | - | **0.2800** | 70.54 | 0.3478 | 0.3899 |
| ACE-Step+SA ControlNet (Hou et al., 2025) | 0.6078 | 0.6336 | 0.6759 | 0.2477 | 110.73 | 0.2479 | 0.3874 |
| ACE-Step+SA ControlNet+LoRA ( Hou et al. (2025)) | 0.6143 | 0.6361 | 0.6741 | 0.2536 | 97.60 | 0.2407 | 0.4114 |
| ACE-Step+MuseControlLite ( Tsai et al. (2025)) | 0.5164 | 0.5275 | 0.6025 | 0.2462 | 68.73 | 0.2764 | 0.4758 |
| SongEcho (Ours) | **0.7066** | **0.7333** | **0.7001** | 0.2674 | **40.37** | **0.2117** | **0.3091** |

In addition, we conduct an experiment involving random swapping of text tags in the test set, with results shown in Table 2. Our method consistently outperforms the other two approaches, with melody-related metrics remaining comparable to those before tag swapping. The CLAP score of our method is 0.0126 lower than that of the original model, which is reasonable since the melody of a song implicitly encodes certain stylistic attributes. This also explains why our method outperforms the original model in terms of CLAP score in Table 1.

Table 3: Quantitative evaluation results on SongEval (Yao et al., 2025).

| | RPA ↑ | RCA ↑ | OA ↑ | CLAP ↑ | FD ↓ | KL ↓ | PER ↓ |
|---|---|---|---|---|---|---|---|
| ACE-Step ( Gong et al. (2025)) | - | - | - | 0.2590 | 71.56 | 0.3305 | 0.4510 |
| ACE-Step+SA ControlNet (Hou et al., 2025) | 0.6463 | 0.6600 | 0.6934 | 0.2666 | 114.18 | 0.4069 | 0.5234 |
| ACE-Step+SA ControlNet+LoRA ( Hou et al. (2025)) | 0.6335 | 0.6465 | 0.6837 | 0.2583 | 104.76 | 0.3112 | 0.5901 |
| ACE-Step+MuseControlLite ( Tsai et al. (2025)) | 0.5421 | 0.5498 | 0.6208 | 0.2600 | 90.19 | 0.3913 | 0.5760 |
| SongEcho (Ours) | **0.7164** | **0.7326** | **0.7097** | **0.2824** | **51.98** | **0.1933** | **0.4487** |

We enhance the annotation of the publicly available SongEval (Yao et al., 2025) benchmark and compare our method with other approaches on this dataset. SongEval comprises 2,399 complete AI-generated songs used to train a song aesthetic evaluation model, exhibiting significant variability in audio quality and lacking lyrics or tag annotations. We select the top 100 English songs with the highest aesthetic scores for testing. Corresponding music tags are generated using Qwen2-audio (Chu et al., 2024). Lyrics transcription files are obtained using Whisper (Radford et al., 2023) combined with All-in-One (Kim & Nam, 2023). Six songs yield text unrecognizable by ACE-Step, resulting in a final test set of 94 songs. The evaluation results, as shown in Table 3, demonstrate that our method achieves superior performance compared with the baselines. The observed decline in PER may result from punctuation errors (e.g., run-on sentences and incorrect sentence breaks) in transcribed lyrics, disrupting their inherent alignment with the melody (see details in Appendix C.2).

**Subjective Evaluation.** For the subjective evaluation, we conduct a Mean Opinion Score (MOS) listening test. Specifically, we randomly select 15 groups, totaling 45 songs, as our evaluation set, each accompanied by the original song and the text prompts for the cover songs. Participants are asked to rate the songs on a scale from 1 to 5 across four dimensions: Melody Fidelity (MF), Text Adherence (TA), Audio Quality (AQ), and Overall Preference (OP). A total of 33 participants, comprising 15 with a music-related background and 18 without, take part in the evaluation. The average scores for each method are shown in Table 4. Our approach achieves the highest scores

Table 4: Mean opinion scores (1–5) comparing melody fidelity (MF), text adherence (TA), Audio Quality (AQ) and overall preference (OP).

| | w/ Music Background | | | | w/o Music Background | | | |
|---|---|---|---|---|---|---|---|---|
| | MF ↑ | TA ↑ | AQ ↑ | OP ↑ | MF ↑ | TA ↑ | AQ ↑ | OP ↑ |
| ACE-Step+SA ControlNet+LoRA (Hou et al., 2025) | 3.056 | 3.285 | 3.085 | 3.104 | 3.133 | 3.636 | 3.182 | 3.160 |
| ACE-Step+MuseControlLite (Tsai et al., 2025) | 2.630 | 3.026 | 2.581 | 2.622 | 2.689 | 3.333 | 2.591 | 2.622 |
| SongEcho (Ours) | **3.644** | **3.800** | **3.756** | **3.819** | **3.884** | **4.160** | **3.916** | **3.942** |

across all four aspects in both groups, demonstrating superior alignment with human perception compared to baselines.

### 5.3.2 QUALITATIVE EVALUATION

The comparison results of our method and other approaches are available at `https://vvanonymousvv.github.io/SongEcho_updated/`. Our model achieves high-quality cover song generation under precise vocal melody control. Overall, the other two methods (Hou et al., 2025; Tsai et al., 2025) can leverage the text control capabilities of ACE-Step to achieve some extent of adaptation, but their audio quality is noticeably inferior to ours. In terms of melody control, ACE-Step+MuseControlLite (Tsai et al., 2025) exhibits noticeable noise and melody drift, along with misalignment between vocals and the original audio, likely due to the cross-attention mechanism's failure to establish precise temporal alignment. ACE-Step+SA ControlNet+LoRA (Hou et al., 2025) achieves decent melody control, but alignment between vocals and melody occasionally falters, and the integration of vocal tracks with the accompaniment lacks coherence. We also provide results on Tag-Melody Conflict, Inpainting & Outpainting, and Global Tempo & Key Control on the demo page. We observe that the model prioritizes the source melody when the provided style tags conflict with the melody condition. Our method supports inpainting and outpainting via a simple masking strategy. Additionally, our method allows direct control over global tempo and key by performing simple post-processing on the extracted vocal melody (F0) sequence. Arbitrary tempo changes are achieved via time-stretching, while key transposition is realized through pitch shifting.

Table 5: Ablation study of our method. "w/ EA" represents replace EiLM with element-wise addition, and "IA-EiLM→Self-Attn" indicates that we insert our IA-EiLM module before the self-attention layer in each Transformer block.

| | RPA ↑ | RCA ↑ | OA ↑ | CLAP ↑ | FD ↓ | KL ↓ | PER ↓ |
|---|---|---|---|---|---|---|---|
| w/ EA, w/o IACR | 0.6336 | 0.6476 | 0.6683 | 0.3014 | 73.83 | 0.1689 | 0.3276 |
| w/ EiLM, w/o IACR | 0.6799 | 0.7000 | 0.6793 | 0.2999 | 75.28 | 0.1569 | 0.3166 |
| IA-EiLM→Self-Attn | 0.6190 | 0.6429 | 0.6303 | 0.3195 | 47.34 | 0.1434 | 0.3462 |
| 100 Training Samples | 0.4677 | 0.4889 | 0.4812 | 0.2854 | 71.85 | 0.1402 | 0.4159 |
| 1000 Training Samples | 0.6505 | 0.6775 | 0.6559 | 0.3115 | 48.59 | 0.1135 | **0.2871** |
| SongEcho (Ours) | **0.7080** | **0.7339** | **0.6952** | **0.3243** | **42.06** | **0.1123** | 0.2951 |

### 5.4 ABLATION STUDY

We conduct a series of ablation experiments to demonstrate the effectiveness of our method. First, we replace our EiLM module with element-wise addition and remove the IACR Module. The results, presented in the 1st and 2nd rows of Table 5, show that the EiLM module improves melody metrics while maintaining comparable performance on other metrics. Building on the 2nd row, incorporating the IACR module yields our final version. The results indicate that the IACR module not only enhances melody metrics but also substantially improves audio quality metrics, underscoring the critical role of adaptively adjusting melody features based on the hidden states of the generative model for harmonious integration of melody conditions.

In our final version, the IA-EiLM module is integrated before the Feed-Forward Network (FFN) layer in each Transformer block. Compared to integrating it before the Self-Attention layer, this

placement results in better melody metrics. This is likely because Self-Attention performs global information interaction, which may disrupt melody preservation, whereas the FFN layer only conducts local transformations, preserving the injected melody features.

We also investigate the impact of training data scale. Training with only 100 samples proves insufficient for effective melody control. However, increasing the training sample size to 1,000 markedly improves performance, with some metrics approaching those achieved with our full dataset. This indicates that our method is highly data-efficient and demonstrates strong potential for application in limited-data scenarios.

## 5.5 DISCUSSION AND LIMITATIONS

Although our method effectively enables song reinterpretation while preserving vocal melodies, the inherent limitations of ACE-Step's text control capabilities (Gong et al., 2025) restrict fine-grained control over vocal timbre, supporting only gender-based adjustments and limiting nuanced voice manipulation. This constraint limits the flexibility of cover song generation, which future advancements in song generation foundation models may address. Alternatively, we plan to integrate a speaker encoder in the future, such as those used in Singing Voice Conversion (SVC), to enable more nuanced and expressive cover song generation.

In this work, we exclude the song-specific, local adaptations (e.g., variations in phoneme durations, vibrato, and note transitions) that musicians may introduce when creating covers. One promising avenue toward Whitney Houston-style expressive covers is to combine our method with melody editing tools or human creators. Local creative modifications can be introduced via external editing or live performance, after which our model generates a global reinterpretation of the revised melody contour. Additionally, AI-generated songs lack the expressive subtlety of human singing and fine-grained vocal technique annotations, preventing our model from achieving the micro-level expressiveness typical of professional covers. Future research could incorporate such fine-grained control by developing song generation models capable of understanding time-aligned musical prompts for precise adaptation control. More ideally, by constructing real paired original-cover datasets, models could learn to autonomously reinterpret an incomplete melody, employing both global and local adaptations to convey distinct emotions and styles.

## 6 CONCLUSION

We introduce a lightweight framework for our cover song generation built upon a text-to-song model. To achieve precise melodic control and harmonious integration, we propose a novel conditioning method, IA-EiLM, which enhances the conditional injection mechanism and conditional representation. The EiLM facilitates temporally aligned modulation of the generative hidden states based on conditioning inputs, while the IACR module employs adaptive refinement, leveraging hidden states to enhance the integration of conditional features into the generative model. Experiments demonstrate that our approach outperforms state-of-the-art melody-controllable music generation methods while requiring significantly fewer trainable parameters. IA-EiLM significantly improves the generation quality and melody preservation for cover songs. Theoretically, IA-EiLM shows potential for application in various conditional tasks beyond controllable music generation. The curated Suno70k dataset helps mitigate copyright issues in song-related AI tasks, supporting advancements in AI music research.

## ETHICS STATEMENT

In exploring cover song generation, we have given full consideration to the ethics of copyright. To mitigate these issues, our model was trained exclusively on AI-generated music. All outputs are strictly for non-commercial academic demonstration, and we are committed to the responsible use of this technology.

## REPRODUCIBILITY STATEMENT

To ensure the reproducibility of our research, we provide a detailed description of the dataset processing pipeline in Section 4. Comprehensive details of computational resources, parameter settings, and evaluation protocols are included in Section 5. Additionally, we release the code, dataset, and audio samples at `https://github.com/lsfhuihuiff/SongEcho_ICLR2026`.

## ACKNOWLEDGMENTS

This work was supported in part by the National Natural Science Foundation of China under Grant No. 62572458, in part by the China Scholarship Council (CSC) under No. 202504910321, in part by the German Research Foundation (DFG) under Germany's Excellence Strategy – EXC 2117 – 422037984, and in part by the National Science and Technology Council, Taiwan under Grant 113-2221-E006-161-MY3.

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

# A    RELATED WORK

## A.1    TEXT-TO-SONG GENERATION

Jukebox (Dhariwal et al., 2020) pioneered the generation of song, utilizing a multiscale Vector Quantized Variational Autoencoder (VQ-VAE) to compress raw audio into discrete codes, which are subsequently modeled using autoregressive Transformers. In recent years, several industry tools, such as Suno[7], Udio[8]), Seed-Music (Bai et al., 2024)), Meruka[9]), have demonstrated promising results in song generation. This progress has spurred researchers to focus on developing open-source text-to-song models, which are increasingly competitive with their closed-source counterparts.

Melodist (Hong et al., 2024) and Melody (Li et al., 2024a) employ a two-stage generation process, sequentially producing vocals and accompaniment to create the final song. Songcreator (Lei et al., 2024)) integrates a dual-sequence language model with a diffusion model to achieve lyrics-to-song generation. Meanwhile, YuE (Yuan et al., 2025)) and SongGen (Liu et al.)) explore the potential of separating vocal and accompaniment tokens in song generation. Building on this, LeVo (Lei et al., 2025) introduces mixed-token generation and leverages Direct Preference Optimization (DPO) to enhance the musicality and instruction-following capabilities of generated songs. A notable limitation of autoregressive model-based approaches is their prolonged inference times. Diffusion-based methods, such as DiffRhythm (Ning et al., 2025) and AceStep (Gong et al., 2025), have significantly mitigated this issue. AceStep (Gong et al., 2025)) further addresses the shortcomings of DiffRhythm (Ning et al., 2025) by adding the understanding of song structure. Although current models generate high-quality songs and some support audio prompts, precise temporal melody control remains challenging. To optimize inference speed and performance, we adopt AceStep as our base model.

## A.2    SING VOICE SYNTHESIS & CONVERSION

Singing Voice Synthesis (SVS) aims to generate single-track vocals consistent with given lyrics and musical scores. Previous GAN-based approaches (Huang et al., 2022; Chunhui et al., 2023) suffer from over-smoothing and unstable training, respectively, compromising the naturalness of synthesized singing. Some methods (Zhang et al., 2022b; 2023b; Cui et al., 2024)) leverage the VITS text-to-speech framework for end-to-end SVS. Choi et al. (Choi & Nam, 2022)) propose an SVS method that only needs audio-and-lyrics-pairs, eliminating the need for duration labels. DiffSinger (Liu et al., 2022)) pioneers the use of diffusion models in SVS, significantly enhancing voice quality. While MuSE-SVS (Kim et al., 2023)) introduces singer and emotion control, TCsinger (Zhang et al., 2024a)) and TCsinger2 (Zhang et al., 2025)) further explore zero-shot style-controllable SVS. Sing Voice Conversion (VC) seeks to transform the timbre and singing techniques of a source singer into those of a target singer while preserving song content and melody. DiffSVC (Liu et al., 2021) applies diffusion models to SVC, improving generation quality, while CoMoSVC (Lu et al., 2024) and LCM-SVC (Chen et al., 2024) focus on accelerating diffusion inference. Additionally, So-VITS-SVC [10]), FreeSVC (Ferreira et al., 2025)), and KNN-SVC (Shao et al., 2025)) achieve zero-shot SVC. While these works delve into melodic control, they remain limited to single-track, short-duration vocal synthesis. In contrast, our task targets the simultaneous generation of full-length accompaniment and vocals.

## A.3    CONTROLLABLE MUSIC GENERATION

Controllable music generation enhances text-to-music generation by integrating temporal control. AirGen (Lin et al., 2024) employs parameter-efficient fine-tuning (PEFT) based on MUSIC-GEN (Copet et al., 2024) for content-based music generation and editing. Li et al. (Li et al., 2024b)) adapt textual inversion (Gal et al., 2023)) into time-varying textual inversion with a bias-reduced stylization technique for example-based style transfer. MusiConGen (Lan et al., 2024)) introduces an in-attention mechanism and efficient fine-tuning to control rhythm and chords. Music Control-

---

[7]https://suno.com/blog/introducing-v4-5

[8]https://www.udio.com/blog/introducing-v1-5

[9]https://www.mureka.ai

[10]https://github.com/svc-develop-team/so-vits-svc

Net (Wu et al., 2024) applies ControlNet (Zhang et al., 2023a) to a diffusion model for text-to-music generation, enabling precise temporal control. Ciranni et al. (Ciranni et al., 2025)) and Hou et al. (Hou et al., 2025)) augment Stable Audio (Evans et al., 2024)), a DiT-based model, with a ControlNet-inspired control branch. However, their reliance on element-wise addition limits control flexibility due to its inherent simplicity. MuseControlLite (Tsai et al., 2025)), inspired by IP-adapter, designs a lightweight adapter for controllable music generation. SongEditor (Yang et al., 2025) uses cross-attention to inject audio conditions, achieving complete vocal or accompaniment tracks when given the rest. However, these cross-attention-based methods require the model to implicitly learn the temporal alignment between the condition and the music tokens. This approach is not only indirect but also incurs significant computational redundancy. Furthermore, existing methods lack adaptive modulation of conditions with original hidden states.

## A.4 Conditional Normalization

Conditional Normalization methods leverage a learned function of conditioning information to derive modulation parameters for affine transformations of target features, proving highly effective across various domains. Conditional Instance Normalization (Dumoulin et al., 2017)) and Adaptive Instance Normalization (AdaIN) (Huang & Belongie, 2017) excel in image style transfer. Conditional Batch Normalization (Anderson et al., 2018) supports general visual question answering, while Dynamic Layer Normalization (Kim et al., 2017) enhances speech recognition. (Perez et al., 2018)) unify these methods with Feature-wise Linear Modulation (FiLM). SPADE (Park et al., 2019)) injects semantic segmentation maps for image translation, and similarly, we propose EiLM for temporally adaptive melody-conditioned control. Temporal FiLM (TFiLM) Birnbaum et al. (2019); Comunità et al. (2023) applies FiLM sequentially within an RNN Elman (1990) framework to capture long-range dependencies, demonstrating robust performance in text classification and black-box audio effect modeling. Recent work DiT (Peebles & Xie, 2023) incorporates conditional information in text-to-image models via the AdaLN-zero module. Despite these advances, the application of conditional normalization in music remains underexplored.

## A.5 Song Dataset

Some of the song datasets for Music Information Retrieval (MIR), such as MIR-1K (Hsu & Jang, 2009), MIR-ST500 (Wang & Jang, 2021), and Cmedia[11], include audio files with rich annotations but are limited to small scales, typically comprising only hundreds of samples. Large-scale song datasets, including WASABI Song Corpus (Buffa et al., 2021), Million Song Dataset (Bertin-Mahieux et al., 2011), and SongCompose-PT (Ding et al., 2024), primarily provide metadata and analytical features but lack raw audio. Datasets designed for singing voice synthesis, such as OpenSinger (Huang et al., 2021), M4singer (Zhang et al., 2022a), and GTsinger (Zhang et al., 2024b), consist of short, single-track vocal segments, making them incompatible with the requirements of Cover Song Generation. Recently, SongEval (Yao et al., 2025) introduced a benchmark dataset of 2,399 AI-generated full songs, but its inconsistent quality limits its applicability.

## B Method Details

### B.1 Note-lyrics Alignment

We condition on the RVMPE-extracted F0 sequence, which is preprocessed by normalizing only its voiced components (50-900Hz) and concatenating the result with a derived binary voiced/unvoiced flag ($uv\_flag$) to form the final melody feature. Our method achieves lyric-to-note alignment without explicit duration modeling or external aligners. The $uv\_flag$ accurately delineates voiced regions, and visualization (see Figure 3) shows that phoneme transitions consistently align with inflection points in the F0 curve. By jointly optimizing melody (F0) and linguistic content (phonemes) during source song reconstruction, the model leverages the strong phoneme-note dependencies captured by its pretrained backbone to implicitly construct a phoneme layout along the F0 timeline.

---

[11]https://www.music-ir.org/mirex/wiki/2020:Singing_Transcription_from_Polyphonic_Music

## B.2 Validity of Equation 6 regarding Audio Copying

Given Eq. 6:

$$(\gamma_m, \beta_m) = \arg\min_{\gamma, \beta} \|E_m(\gamma \odot h + \beta) - M_c\|_2^2, \quad (14)$$

where both $\gamma$ and $\beta$ are static (i.e., independent of the hidden states $h$).

**Well-constrained case** ($\gamma_m = 0$). The objective simplifies to $\min_\beta \|E_m(\beta) - M_c\|_2^2$, which has a unique solution for any $h$.

**Underconstrained case** ($\gamma_m \neq 0$). A static $(\gamma, \beta)$ must satisfy $E_m(\gamma \odot h + \beta) = M_c$ simultaneously for all possible hidden states $h$. This is impossible unless $h$ degenerates to a constant vector.

**Why $\gamma_m = 0$ works for full-audio conditioning** . When the condition $m$ is the full target audio (i.e., the modeling objective effectively becomes reconstructing $m$ via $\gamma \odot h + \beta = m$), the optimal solution to Eq. 6 is

$$\gamma_m \approx \mathbf{0}, \quad \beta_m \approx m, \quad (15)$$

which completely suppresses the hidden state $h$ and directly copies the condition. This is exactly what MuseControlLite does, as confirmed by its diagonal attention pattern under full audio conditioning (see Fig. 2 in the MuseControlLite Tsai et al. (2025) appendix or Fig. 5 in our appendix). When the attention matrix is always diagonal, the query degenerates into a pure positional index and suppresses $h$. The output of the attention layer then becomes:

$$\text{Output} = \text{Softmax}\left(\frac{Q_h K_c^\top}{\sqrt{d}}\right) V_c \approx I \cdot V_c = V_c, \quad (16)$$

where $Q_h$ denotes the query from hidden state $h$, and $K_c$, $V_c$ are derived from the audio condition $m$. This process directly duplicates $V_c$ rather than generating new content.

**Why $\gamma_m = 0$ fails for melody control** In our task, the condition $m$ corresponds to a compressed melody rather than the target latent. Setting $\gamma_m = 0$ causes the modulated hidden states to contain only melody information. They lose essential attributes of the target song (e.g., timbre and lyrics), making it impossible to generate the target song. This establishes that $\gamma_m \neq 0$ is necessary for our task. However, when $\gamma_m \neq 0$, with $\gamma$ and $\beta$ fixed across all hidden states $h$, Eq. 6 becomes underconstrained for arbitrary $h$. Our proposed IACR strategy is introduced specifically to make the objective well-constrained again.

In summary, Audio copying succeeds with static conditioning only because it can exploit the degenerate $\gamma_m = 0$ solution—a shortcut not available for melody control. Our IACR resolves the underconstrained problem by making $(\gamma, \beta)$ dependent on both $m$ and $h$ (Eq. 7), yielding obvious improvement when ablated (Table 5, Row 2 vs. Row 6).

## C Additional Experimental Details

### C.1 Details of Baselines

**ACE-Step+SA ControlNet (Hou et al., 2025; Gong et al., 2025).** We follow the architecture of SA-ControlNet, while replacing its generative backbone with the pre-trained ACE-Step model. Specifically, we clone half of the pre-trained Diffusion Transformer (DiT) blocks to create a control branch consisting of 12 Transformer blocks in total, while keeping the melody encoder exactly the same as in our proposed method. The model is trained with the same training settings as our model (AdamW, $\beta_1=0.9$, $\beta_2=0.95$, weight decay=0.01, lr=$10^{-4}$, 1,000-step linear warm-up). Training lasts for 32,000 steps with a batch size of 12 (vs. 30,000 steps for our model). During inference, we directly adopt the original classifier-free guidance (CFG) sampler of ACE-Step with guidance scale $\lambda = 15.0$.

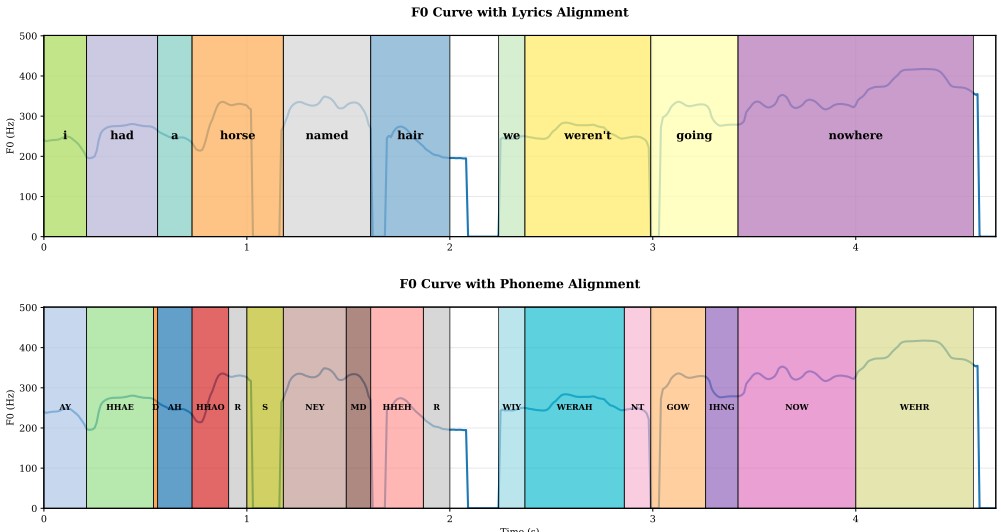

Figure 3: We visualize the F0 contour extracted from the song, along with the word-level and phoneme-level timestamps produced by the Montreal Forced Aligner (MFA) (McAuliffe et al., 2017). The full lyrics used in the example are: "I had a horse named Hair, we weren't going nowhere."

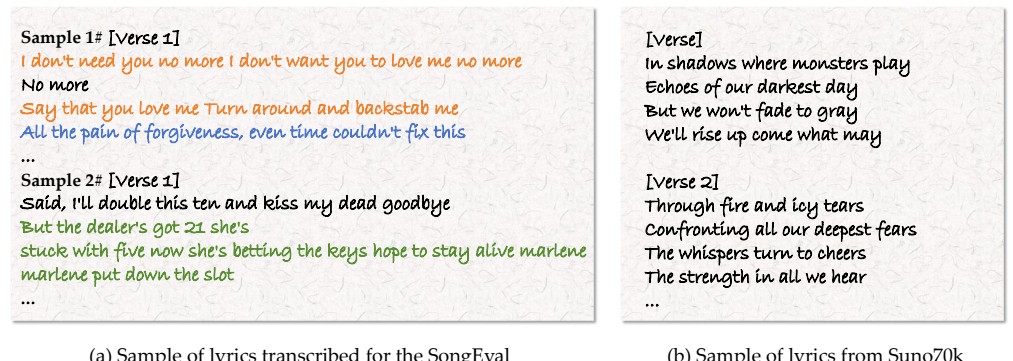

(a) Sample of lyrics transcribed for the SongEval  (b) Sample of lyrics from Suno70k

Figure 4: Lyrics transcribed by Whisper (Radford et al., 2023) with All-in-One (Yao et al., 2025) for SongEval (Yao et al., 2025) exhibit punctuation errors, including run-on sentences (orange), multiple clauses per line (blue), and incorrect sentence breaks (green), whereas Suno70k's native lyrics ensure each phrase is on a separate line.

**ACE-Step+MuseControlLite (Tsai et al., 2025; Gong et al., 2025).** We follow the architecture of MuseControlLite, while replacing its generative backbone with the pre-trained ACE-Step model. Specifically, we adopt its decoupled cross-attention mechanism equipped with Rotary Position Embedding (RoPE), while keeping the melody encoder identical to that used in our proposed method. The model is trained with the same training settings as our model (AdamW, $\beta_1$=0.9, $\beta_2$=0.95, weight decay=0.01, lr=$10^{-4}$, 1,000-step linear warm-up). Training is performed for 36,000 steps with a batch size of 12 (vs. 30,000 steps for our model). During inference, we evaluate three guidance strategies: (1) the original ACE-Step classifier-free guidance (CFG) with guidance scale $\lambda$=15.0; (2) MuseControlLite-style Multiple Classifier-Free Guidance with $\lambda_{\text{text}}$=15.0 and $\lambda_{\text{melody}}$=7.5; (3) Multiple Classifier-Free Guidance with $\lambda_{\text{text}}$=7.5 and $\lambda_{\text{melody}}$=15.0. The first strategy (ACE-Step's native CFG, $\lambda$=15.0) yields the best overall performance. Results for the three variants are summarized in Table 6. The first version achieves the best overall performance and is reported in our main text.

Table 6: Quantitative evaluation results of three inference guidance strategies for the ACE-Step+MuseControlLite baseline (Tsai et al., 2025).

| | | RPA ↑ | RCA ↑ | OA ↑ | CLAP ↑ | FD ↓ | KL ↓ | PER ↓ |
|---|---|---|---|---|---|---|---|---|
| Suno70k | $\lambda = 15.0$ | 0.5205 | 0.5346 | 0.5940 | 0.2977 | 72.04 | 0.2151 | **0.4194** |
| | $\lambda_{text} = 15.0, \lambda_{melody} = 7.0$ | 0.4896 | 0.5042 | 0.5557 | **0.3159** | 55.82 | **0.1820** | 0.4837 |
| | $\lambda_{text} = 7.0, \lambda_{melody} = 15.0$ | **0.5536** | **0.5653** | **0.6351** | 0.2942 | 73.59 | 0.2763 | 0.6159 |
| Suno70k+Swapped Tags | $\lambda = 15.0$ | 0.5164 | 0.5275 | 0.6025 | 0.2462 | 68.73 | 0.2764 | 0.4758 |
| | $\lambda_{text} = 15.0, \lambda_{melody} = 7.0$ | 0.4798 | 0.4924 | 0.5669 | **0.2704** | **54.11** | 0.3087 | 0.5446 |
| | $\lambda_{text} = 7.0, \lambda_{melody} = 15.0$ | **0.5578** | **0.5682** | **0.6459** | 0.2290 | 74.39 | 0.3445 | 0.6387 |
| SongEval | $\lambda = 15.0$ | 0.5421 | 0.5498 | 0.6208 | 0.2600 | 90.19 | 0.3913 | **0.5760** |
| | $\lambda_{text} = 15.0, \lambda_{melody} = 7.0$ | 0.5040 | 0.5115 | 0.5800 | **0.2699** | **77.36** | **0.3205** | 0.6333 |
| | $\lambda_{text} = 7.0, \lambda_{melody} = 15.0$ | **0.5880** | **0.5954** | **0.6614** | 0.2567 | 97.44 | 0.4593 | 0.7179 |

Table 7: Comparison of SongEval (Yao et al., 2025) Aesthetics Metrics Across Methods.

| Dataset | Suno70k | | | | | SongEval | | | | |
|---|---|---|---|---|---|---|---|---|---|---|
| Metrics | Coherence ↑ | Musicality ↑ | Memorability ↑ | Clarity ↑ | Naturalness ↑ | Coherence ↑ | Musicality ↑ | Memorability ↑ | Clarity ↑ | Naturalness ↑ |
| ACE-Step | 3.144 | 2.873 | 2.991 | 2.957 | 2.817 | 3.389 | 3.149 | 3.254 | 3.180 | 3.059 |
| SA ControlNet | 3.237 | 2.958 | 3.058 | 3.018 | 2.909 | 3.317 | 3.061 | 3.131 | 3.072 | 2.953 |
| MuseControlLite | 2.871 | 2.591 | 2.667 | 2.607 | 2.592 | 2.914 | 2.686 | 2.705 | 2.648 | 2.617 |
| SongEcho (Ours) | **3.776** | **3.485** | **3.644** | **3.534** | **3.440** | **3.941** | **3.698** | **3.834** | **3.680** | **3.590** |

## C.2 COMPARISON OF LYRICS FROM SONGEVAL AND SUNO70K

Lyrics transcribed by Whisper (Radford et al., 2023) with All-in-One (Yao et al., 2025) for SongEval (Yao et al., 2025) exhibit punctuation errors, including run-on sentences, multiple clauses per line, and incorrect sentence breaks, due to inaccurate segment splitting by Whisper. In contrast, Suno70k's native lyrics ensure each phrase occupies a separate line (See Fig. 4). During training, this implicitly fosters alignment between lyrics and melody, with each melodic phrase corresponding to one lyric line. However, transcribed lyrics disrupt this alignment, leading to increased PER in the SongEval dataset evaluation. The lesser impact on the original model stems from its lack of melody control, as it generates lyrics sequentially without requiring lyric-melody alignment, thus relying less on accurate sentence segmentation.

## C.3 AESTHETIC EVALUATION

We perform an aesthetic evaluation of the results from two datasets using SongEval (Yao et al., 2025), as shown in Table 7. Each song is evaluated across five dimensions: overall coherence, memorability, naturalness of vocal breathing and phrasing, clarity of song structure, and overall musicality. Our method exhibits a clear advantage over other approaches across all metrics, further validating the harmonious integration of melody control and the generative model, thereby generating high-aesthetic cover songs.

## D THE USE OF LARGE LANGUAGE MODELS (LLMS)

We used large language models (LLMs) as a general-purpose writing assistant to polish the text of this paper. Specifically, LLMs were employed to correct grammar, improve clarity, and refine phrasing in certain sentences. The models did not contribute to the research design, problem formulation, method development, experimentation, interpretation of results, or overall scientific contributions. Their role was limited solely to surface-level editing and presentation improvements of the manuscript.

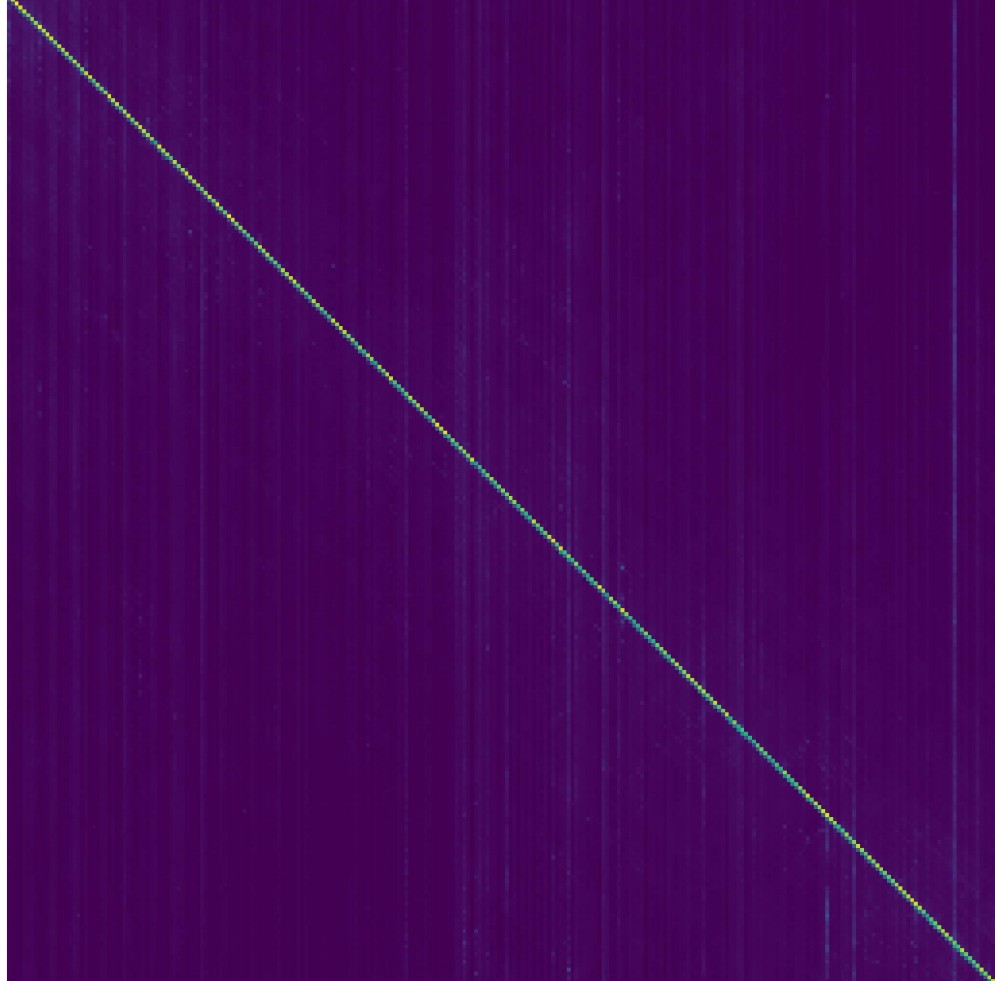

Figure 5: Attention map visualization of MuseControlLite (Tsai et al., 2025) under full-audio conditioning. The clear diagonal pattern indicates that the post-softmax attention matrix approximates an identity matrix.

