# OpenReview forum: "SongEcho: Towards Cover Song Generation via Instance-Adaptive Element-wise Linear Modulation"
_ICLR.cc/2026/Conference — ICLR 2026 Poster_

### Official Review · Reviewer_dAnD · 2025-10-19

**Soundness:** 3
**Presentation:** 2
**Contribution:** 3
**Rating:** 4
**Confidence:** 5

**Summary:**

The paper formalizes Cover Song Generation, preserving the vocal melody while generating a new song. The authors compare three methods for controllable music (song) generation: ControlNet, MuseControlLite, and their proposed model, SongEcho. Through multiple experiments, they show that SongEcho outperforms the others in terms of music quality, text adherence, and vocal melody preservation.

**Strengths:**

1. By using fewer parameters to adapt Acestep to the Cover Song Generation task, it demonstrates better performance compared to other fine-tuning methods, showing the effectiveness of the proposed approach.
2. Provides a critical analysis across different conditioning mechanisms.
3. The authors promised to open-source a dataset enriched with detailed annotations, including enhanced tags and lyrics.

**Weaknesses:**

1. The authors adopt two baseline methods that are not originally applied to Acestep. The authors should provide implementation details for these baseline methods (e.g., training specifics, inference specifics) to ensure that both the baseline methods and the proposed method are treated equally.
2. The original baseline methods might not be suitable for Acestep. ControlNet can be applied by simply copying parts of Acestep; however, regarding the linear attention architecture, MuseControlLite might require modifications since it uses the traditional cross-attention method with positional encoding and zero-convolution. There might be a misalignment if the cross-attention layers in Acestep use linear attention, but the decoupled cross-attention in MuseControlLite are not. Additionally, the original MuseControlLite paper indicates that during inference, multiple classifier-free guidance is necessary. The authors should provide details for both the training and inference processes for the baseline methods.
3. The authors indicate that the ControlNet method is not fully trainable, but only the LoRA adapters are trainable. This makes the comparison less persuasive; perhaps the authors should try mixed-precision training.
4. The authors define 'Cover Song Generation' as 'preserving the source vocal melody while simultaneously synthesizing new vocals and accompaniment'; however, generating a song with the exact same vocal melody might not fully represent 'cover song' generation. A 'cover song' does not mean that the vocal melody has to be exactly the same.

**Questions:**

1.  Can the authors provide more details about the training and inference specifics?
2.  I understand that the formulation of 'Cover Song Generation' limits the definition to 'using the same vocal melody and regenerating it,' which might be easier. However, a cover song traditionally implies that the musical elements, such as instrumentation, tempo, or melodic phrasing, might change, while the fundamental character of the song remains the same. How will you address your work in light of this definition? Otherwise, the work might be more closely related to a vocal-to-accompaniment generation task.
3. The authors claim that "cross-attention mechanism" are redundant. However, in the MuseControlLite paper, they claim that this is for the use of improvisation (i.e. if only partial condition is provided, the model can improvise for the rest), does songecho support this?

This is an interesting paper, I am willing to raise the score if answered properly.

---

> ### Author Response · Authors · 2025-11-23
> **Response to Reviewer dAnD (Part 1/3)**
>
> We are grateful for your review and valuable comments, and we hope our response fully resolves your concerns. Changes are highlighted in the updated manuscript.
> ### Q1. Training and Inference Details of Baselines
> We add training and inference details of baselines in Appendix C.1.
>
> **[ACE-Step+SA ControlNet]** (Line 909-916)
> "We follow the architecture of SA-ControlNet, while replacing its generative backbone with the pre-trained ACE-Step model. Specifically, we clone half of the pre-trained Diffusion Transformer (DiT) blocks to create a control branch consisting of 12 Transformer blocks in total, while keeping the melody encoder exactly the same as in our proposed method. The model is trained with the same training settings as our model (AdamW, β₁=0.9, β₂=0.95, weight decay=0.01, lr=10⁻⁴, 1,000-step linear warm-up). Training lasts for 32,000 steps with a batch size of 12 (vs. 30,000 steps for our model). During inference, we directly adopt the original classifier-free guidance (CFG) sampler of ACE-Step with default guidance scale λ=15.0."
>
> **[ACE-Step+MuseControlLite]** (Line 961-971)
> "We follow the architecture of MuseControlLite, while replacing its generative backbone with the pre-trained ACE-Step model. Specifically, we adopt its decoupled cross-attention mechanism equipped with Rotary Position Embedding (RoPE), while keeping the melody encoder identical to that used in our proposed method. The model is trained with the same training settings as our model (AdamW, β₁=0.9, β₂=0.95, weight decay=0.01, lr=10⁻⁴, 1,000-step linear warm-up). Training is performed for 36,000 steps with a batch size of 12 (vs. 30,000 steps for our model). During inference, we evaluate three guidance strategies: (1) the original ACE-Step classifier-free guidance (CFG) with guidance scale $λ=15.0$; (2) MuseControlLite-style Multiple Classifier-Free Guidance with $λ_{text}=15.0$ and $λ_{melody}=7.5$; (3) Multiple Classifier-Free Guidance with $λ_{text}=7.5$ and $λ_{melody}=15.0$. The first strategy (ACE-Step's native CFG, $λ=15.0$) yields the best overall performance. Results for the three variants are summarized in Table 6. The first version achieves the best overall performance and is reported in our main text."
>
> ### Q2. Multiple Classifier-Free Guidance of MuseControlLite
> We add the inference results for ACE-Step+MuseControlLite using Multiple Classifier-Free Guidance (MCFG). As shown in Table 1 (Table 6 in the updated manuscript), we observe a trade-off between text adherence and melody control: increasing the text guidance scale improves song-tag alignment (CLAP) but compromises melody preservation (RPA, RCA, OA), whereas increasing the melody guidance scale enhances melody fidelity at the expense of song-tag alignment. The first version achieves the best overall performance and is reported in our main text.
>
> Table 1: Quantitative evaluation of three inference guidance strategies on the ACE-Step+MuseControlLite baseline. The best results are highlighted in **bold**, and the second-best results are shown in *italic*.
> | Dataset                  | λ setting                              | RPA ↑       | RCA ↑       | OA ↑        | CLAP↑      | FD ↓       | KL ↓        | PER ↓       |
> |--------------------------|----------------------------------------|-------------|-------------|-------------|-------------|------------|-------------|-------------|
> | **Suno70k**              | $λ=15.0$                                 | _0.5205_    | _0.5346_    | _0.5940_    | _0.2977_    | _72.04_    | _0.2151_    | **0.4194**  |
> |                          | $λ_{text}=15.0, λ_{melody}=7.0$              | 0.4896      | 0.5042      | 0.5557      | **0.3159**  | **55.82**  | **0.1820**  | _0.4837_    |
> |                          | $λ_{text}=7.0, λ_{melody}=15.0$              | **0.5536**  | **0.5653**  | **0.6351**  | 0.2942      | 73.59      | 0.2763      | 0.6159      |
> | **Suno70k+Swapped Tags** | $λ=15.0$                                | _0.5164_    | _0.5275_    | _0.6025_    | _0.2462_    | _68.73_    | **0.2764**  | **0.4758**  |
> |                          | $λ_{text}=15.0, λ_{melody}=7.0$              | 0.4798      | 0.4924      | 0.5669      | **0.2704**  | **54.11**  | _0.3087_    | _0.5446_    |
> |                          | $λ_{text}=7.0, λ_{melody}=15.0$              | **0.5578**  | **0.5682**  | **0.6459**  | 0.2290      | 74.39      | 0.3445      | 0.6387      |
> | **SongEval**             | $λ=15.0$                                 | _0.5421_    | _0.5498_    | _0.6208_    | _0.2600_    | _90.19_    | _0.3913_    | **0.5760**  |
> |                          | $λ_{text}=15.0, λ_{melody}=7.0$              | 0.5040      | 0.5115      | 0.5800      | **0.2699**  | **77.36**  | **0.3205**  | _0.6333_    |
> |                          | $λ_{text}=7.0, λ_{melody}=15.0$              | **0.5880**  | **0.5954**  | **0.6614**  | 0.2567      | 97.44      | 0.4593      | 0.7179      |

---

> > ### Author Response · Authors · 2025-11-23
> > **Response to Reviewer dAnD (Part 2/3)**
> >
> > ### Q3. Adaptation of MuseControlLite to ACE-Step
> > In the ACE-Step architecture, linear attention is applied exclusively to the self-attention mechanism (see Fig.1 in ACE-Step [1]), while the cross-attention layers used for condition injection retain the traditional attention calculation method. Consequently, MuseControlLite can be seamlessly integrated with the ACE-Step backbone. We also maintain the use of zero-convolution and RoPE positional embeddings as the original MuseControlLite design to ensure a fair comparison.
> >
> > ### Q4. Mixed Precision Training for ACE-Step + SA ControlNet
> > We conduct full-parameter training on the control branch of ACE-Step + SA ControlNet (totaling 1.6B trainable parameters, roughly 33x the size of ours) using bf16 mixed precision. As shown in Table 2, although each version presents specific strengths and weaknesses, they demonstrate comparable overall performance. Thus, our method consistently outperforms the ControlNet-based baseline across all metrics. Results of the full-parameter training variant are added to the updated manuscript (Table 1,2,3 in the updated version).
> > Table 2: Quantitative evaluation of two training strategies on the ACE-Step+ControlNet baseline.
> > | Dataset                  | Method   | RPA ↑    | RCA ↑    | OA ↑     | CLAP↑     | FD ↓    | KL ↓     | PER ↓    | TP ↓   |
> > |--------------------------|----------|----------|----------|----------|------------|---------|----------|----------|--------|
> > | **Suno70k**              | w/o LoRA | 0.6209   | 0.6440   | 0.6858   | 0.2875     | 105.95  | 0.2019   | 0.3714   | 1.6B   |
> > |                          | w/ LoRA  | 0.6214   | 0.6431   | 0.6833   | 0.2892     | 99.19   | 0.1850   | 0.3734   | 331M   |
> > | **Suno70k+Swapped Tags** | w/o LoRA | 0.6078   | 0.6336   | 0.6759   | 0.2477     | 110.73  | 0.2479   | 0.3874   | -      |
> > |                          | w/ LoRA  | 0.6143   | 0.6361   | 0.6741   | 0.2536     | 97.60   | 0.2407   | 0.4114   | -      |
> > | **SongEval**             | w/o LoRA | 0.6463   | 0.6600   | 0.6934   | 0.2666     | 114.18  | 0.4069   | 0.5234   | -      |
> > |                          | w/ LoRA  | 0.6335   | 0.6465   | 0.6837   | 0.2583     | 104.76  | 0.3112   | 0.5901   | -      |
> >
> > ### Q5. Task Definition and Clarification
> > We agree with the reviewer that 'cover song' in a broad artistic sense involves local changes in rhythm or melodic phrasing.
> > **[Ambiguity of "Vocal-to-accompaniment generation"]**
> > However, "Vocal-to-accompaniment generation" risks ambiguity with existing tasks such as "Vocal-to-accompaniment synthesis" in Melodist [2], "Singing-to-accompaniment" in ACE-Step [1], and "Vocal-to-song" in SongEditor [3]. These prior methods generate only an accompaniment track from an input vocal track. In contrast, our input consists of a melody sequence, lyrics and tags, from which we simultaneously generate new vocals and accompaniment.
> >
> > **[Clarification of Task Definition]**
> > Inspired by image style transfer [4], which reformulates the painting process as texture synthesis while strictly preserving content structure and explicitly ignoring structural deformations in real paintings, we adopt an analogous paradigm for cover song generation.
> > We retain the term "Cover Song Generation" and explicitly clarify the task scope in the "Introduction" and "Limitations" sections that our task focuses on global reinterpretations while disregarding customized local adaptations. (Our global tempo and key adaptations are demonstrated in Section 5 of the [updated demo](https://vvanonymousvv.github.io/SongEcho_updated/).)
> >
> > **[Revised version]**
> > The specific revisions are as follows:
> > **[Introduction]** (Line 44-48, Task definition and scope):
> > "Similar to Whitney Houston's rendition, musicians creating cover songs may introduce flexible adaptations in local musical elements, such as phoneme durations, vibrato, and note transitions, yielding highly varied and personalized reinterpretations.
> > In this work, we abstract a cover paradigm applicable to arbitrary songs and reformulate our cover song generation as a conditional generation task that performs a global style transfer with text guidance while preserving the source vocal melody contour and disregarding local customized adaptations."
> >
> > **[Limitations]** (Line 498-499, 505-509, Task scope and future directions)
> > "In this work, we exclude the song-specific, localized modifications (e.g., variations in phoneme durations, vibrato, and note transitions) that musicians may introduce when creating covers. ... Future research could incorporate such fine-grained control by developing song generation models capable of understanding time-aligned musical prompts for precise adaptation control. More ideally, by constructing real paired original-cover datasets, models could learn to autonomously reinterpret an incomplete melody, employing both global and local adaptations to convey distinct emotions and styles."

---

> > > ### Author Response · Authors · 2025-11-23
> > > **Response to Reviewer dAnD (Part 3/3)**
> > >
> > > ### Q6. Support for Improvisation
> > > Our method supports improvisation as shown in Section 4 of the [updated demo](https://vvanonymousvv.github.io/SongEcho_updated/). Like MuseControlLite, we perform both inpainting and outpainting via a simple masking strategy.
> > >
> > > Thank you again for the constructive feedback. We are happy to address any further questions or concerns you may have.
> > >
> > > [1] ACE-Step: A Step Towards Music Generation Foundation Model. 2025
> > > [2] Text-to-Song: Towards Controllable Music Generation Incorporating Vocals and Accompaniment. ACL 2024
> > > [3] SongEditor: Adapting Zero-Shot Song Generation Language Model as a Multi-Task Editor. AAAI 2025
> > > [4] Arbitrary Style Transfer in Real-Time with Adaptive Instance Normalization. ICCV 2017

---

> > > > ### Comment · Reviewer_dAnD · 2025-11-24
> > > >
> > > > Thanks for the reply and the comprehensive experiments.
> > > > I have a few more questions.
> > > > 1. Can you share how you calculate the trainable parameter count (e.g. show the details of each layer)? As I look into the architecture of MuseControlLite and Soug Echo, I found that the number of trainable layers in every DiT block is similar:
> > > >     - MuseControlLite: 2 linear layers + 1 zero-convolution layer
> > > >     - SongEcho: 3 linear layers
> > > >    - I am not sure why there is a gap between the number of trainable parameters.
> > > > 2. What is the difference between EiLM and FiLM? It seems that, except for the shape of the injective condition, it looks the same.
> > > > 3. In the paper:
> > > > >Without access to the hidden states h or their intrinsic melody Mh, the transformation network T must learn a universal mapping ∆Mh→Mc across all possible h, causing Equation 6 to be underconstrained.
> > > >
> > > > Actually, I doubt that. When I try MuseControlLite with the audio condition, which directly copies the reference audio, the copied audio sounds exactly the same as the reference audio. If "Equation 6" stands, MuseControlLite will not be able to copy the reference audio without an IACR-like design.

---

> > > > > ### Author Response · Authors · 2025-11-26
> > > > > **Response to Reviewer dAnD (Part1/2)**
> > > > >
> > > > > We are grateful for your valuable questions and hope our responses fully address them.
> > > > > ### Q1. Calculation of Trainable Parameters
> > > > > The disparity in parameter counts arises from the specific dimension settings within the model layers.
> > > > >
> > > > > **[MuseControlLite Configuration]**
> > > > > The specific configuration for each DiT block in MuseControlLite is:
> > > > > * `melody_to_k`: `nn.Linear(256, 2560, bias=False)`
> > > > > * `melody_to_v`: `nn.Linear(256, 2560, bias=False)`
> > > > > * `conv_out`: `nn.Conv1d(2560, 2560, kernel_size=1, bias=False)`
> > > > >
> > > > > **Total Parameters:**
> > > > > With a melody encoder size of $P_{m} = 0.33\text{M}$, the calculation is:
> > > > > $$P_{total} = 0.33\text{M} + \left[ (256 \times 2560) \times 2 + (2560 \times 2560) \right] \times 24 \approx 189\text{M}$$
> > > > >
> > > > > **[SongEcho Configuration (Ours)]**
> > > > > The specific configuration for each DiT block in SongEcho is:
> > > > > * `proj_c`: `nn.Linear(256, 256)`
> > > > > * `proj_x`: `nn.Linear(2560, 256)`
> > > > > * `cond_project`: `nn.Linear(256, 2560 * 2, bias=False)`
> > > > >
> > > > > **Total Parameters:**
> > > > > $$P_{total} = 0.33\text{M} + \left[ (256+1) \times 256 + (2560+1) \times 256 + (256 \times 2560 \times 2) \right] \times 24 \approx 49.1\text{M}$$
> > > > >
> > > > > ### Q2. Novelty and Distinction of EiLM
> > > > > We acknowledge that EiLM shares the core affine transformation formula ($y = \gamma \odot x + \beta$) with FiLM. However, they fundamentally differ in **mechanism** and **application scope**:
> > > > >
> > > > > **[Mechanism: Global Time-Invariant vs. Local Time-Varying Conditioning]**
> > > > > Standard FiLM applies **time-invariant** parameters (broadcasting across time), acting as a global constraint. In contrast, EiLM generates **time-varying** parameters $(\gamma_t, \beta_t)$. This structurally shifts the mechanism from global style injection to local, fine-grained temporal control, representing an elegant and necessary extension for sequential conditioning.
> > > > >
> > > > > **[Novel Application]**
> > > > > Standard FiLM cannot be directly applied to our task. To our knowledge, EiLM marks the first application of FiLM-style modulation to temporally controllable music generation. It offers an alternative to Cross-Attention and Element-wise addition, balancing computational efficiency with modeling flexibility.
> > > > >
> > > > > **[Effectiveness]**
> > > > > Ablation studies further demonstrate that EiLM effectively enhances melody controllability (Table 5, Rows 1 and 2).

---

> > > > > > ### Author Response · Authors · 2025-11-26
> > > > > > **Response to Reviewer dAnD (Part2/2)**
> > > > > >
> > > > > > ### Q3. Validity of Equation 6 regarding Audio Copying
> > > > > >
> > > > > > We thank the reviewer for this insightful question.
> > > > > > MuseControlLite’s near-perfect audio copying does not contradict our claim that static conditioning (Eq. 6) is underconstrained. It is a degenerate special case unavailable in melody-controllable generation.
> > > > > >
> > > > > > **[If $\gamma_m=0$, Eq.6 is well-constrained]**
> > > > > >
> > > > > > Given Eq.6: $(\gamma_{m}, \beta_{m}) = \arg\min_{\gamma, \beta} || E_m(\gamma \odot {h} + \beta) - M_c ||_{2}^{2}$, where both $\gamma$ and $\beta$ are static (*i.e.*, independent of the hidden states $h$).
> > > > > >
> > > > > > ***Well-constrained case***:
> > > > > > When $\gamma_m = \mathbf{0}$, the objective reduces to $\min_{\beta}||E_m(\beta) - M_c||_2^2$, which has a unique solution for any $h$.
> > > > > >
> > > > > > ***Underconstrained case:***
> > > > > > When $\gamma_m \neq \mathbf{0}$, static $(\gamma, \beta)$ must satisfy $E_m(\gamma \odot h + \beta) = M_c$ simultaneously for all possible $h$. This is impossible unless $h$ degrades to a constant.
> > > > > >
> > > > > > **[$\gamma=0$ works in audio control]**
> > > > > > When the condition $m$ is the full target audio (*i.e.*, the modelling objective effectively becomes reconstructing $m$ via $\gamma \odot h + \beta = m$), the optimal solution to Equation (6) is as follows:
> > > > > > $$\gamma_m \approx \mathbf{0}, \quad \beta_m \approx m,$$
> > > > > > which completely suppresses the hidden state $h$ and directly copies the condition. This is exactly what MuseControlLite does, as confirmed by its diagonal attention pattern under full audio conditioning (see Fig. 2 in the MuseControlLite[1] appendix or Fig. 5 in our appendix). When the attention matrix is always diagonal, the query degenerates into a pure positional index and suppresses $h$. The output of the attention layer then becomes:
> > > > > > $$\text{Output} = \mathrm{Softmax}\left(\frac{Q_h K_c^\top}{\sqrt{d}}\right) V_c  \approx I \cdot V_c = V_c,$$
> > > > > > where $Q_h$ denotes the Query from hidden state $h$, $K_c$ and $V_c$ are derived from the audio condition $m$. This process directly duplicates $V_c$ rather than generating new content.
> > > > > >
> > > > > >
> > > > > > **[$\gamma=0$ fails for vocal melody control]**
> > > > > > In contrast, when the condition $m$ corresponds to a compressed melody rather than the target latent, $\gamma_m = 0$ causes the modulated hidden states to contain only melody information. They lose essential attributes of the target song (e.g., timbre and lyrics), making it impossible to generate the target song. This establishes that $\gamma_m \neq 0$ is necessary for our task. However, when $\gamma_m \neq 0$, with $\gamma$ and $\beta$ fixed across all hidden states $h$, Eq.6 becomes unconstrained for arbitrary $h$. Our proposed IACR strategy is introduced specifically to make the objective well-constrained again.
> > > > > >
> > > > > > **[Conclusion]**
> > > > > > Audio copying succeeds with static conditioning only because it can exploit the degenerate $\gamma_m = 0$ solution, a shortcut not available for melody control. Our IACR resolves the underconstrained problem by making $ (\gamma, \beta) $ dependent on both $m$ and $h$ (Eq. 7), yielding obvious improvement when ablated (Table 5, Row 2 vs. Row 6).
> > > > > >
> > > > > > We now explicitly state that $\gamma_m \neq \mathbf{0}$ in the melody control setting (Line 251 of the revised manuscript).
> > > > > >
> > > > > > [1] MuseControlLite: Multifunctional Music Generation with Lightweight Conditioners. ICML 2025

---

> > > > > > > ### Comment · Reviewer_dAnD · 2025-11-27
> > > > > > >
> > > > > > > Nice catch! The authors reply fully answered my questions, I raised my score.

---

> > > > > > > > ### Author Response · Authors · 2025-11-27
> > > > > > > > **Thank you**
> > > > > > > >
> > > > > > > > Thank you! We are glad that our response addressed your questions. We truly appreciate your time and the raised score.

---

### Official Review · Reviewer_5LhQ · 2025-10-28

**Soundness:** 3
**Presentation:** 2
**Contribution:** 3
**Rating:** 2
**Confidence:** 4

**Summary:**

The paper presents a novel approach for what is described as cover song generation, but should better be denoted as music generation with melody conditioning. The model takes as input the lyrics, a reference melody, and other control features, and generates both singing voice and accompaniment audio. It is built by means of adapting the foundational model ACE-Step. Instance-Adaptive Element-wise Linear Modulation (IA-EiLM) for temporal conditioning, together with Instance-Adaptive Condition Refinement (IACR) to inject conditioning information during diffusion. The proposed model enables coordinated generation of vocals and accompaniment in a stylistically consistent and coherent manner. To support further research in this area, the authors propose a new dataset of synthetic music collected from Suno, denoted Suno70k.

**Strengths:**

- adaptation of a pretrained linear DiT model to a new control, establishing a new approach for melody conditioning of music generation
- two rebranded approaches EiLM and ICAR for conditioning
- a new large dataset of full music, including singing voice and lyrics annotations.
- successful evaluation of the proposed methods.

**Weaknesses:**

The style of presentation is rather unclear and confusing:

- in the abstract the authors introduce the term cover song generation as: (Line 17) *We formalize this challenge as Cover Song Generation, which requires preserving the source vocal melody while simultaneously synthesizing new vocals and accompaniment, posing higher demands for controllable music generation.* they later change into:  (line 40) *...reinterpret the original’s emotional and stylistic core, evolving a gentle country ballad into a worldwide anthem of deep affection. Such reinterpretations amplify a song’s cultural impact and illustrate the creative potential of musical reimagination.*

I would stress that the Whitney Houston example demonstrates exactly what the proposed model cannot achieve. Whitney Houston alters phoneme durations, vibrato, intensity, and note transitions to modify the emotional expressivity. Conditioning the model on a fixed melodic contour from the original song is incompatible with this goal. There are clearly ways to use melody conditioning to achieve some sort of reinterpretation. It would be helpful to briefly describe how this could be achieved. (See questions.)

___

- Line 093: *Feature-wise Linear Modulation (FiLM) (Perez et al., 2018) has demonstrated efficacy as a conditioning technique. However, FiLM is limited to injecting global conditions, uniformly modulating all tokens within an instance, making it unsuitable for time-varying features like melodies.*

It is unclear to me how this claim can be made. While it is true that FiLM is most of the time used for global features, there is nothing in the technique that prevents one from using it to represent time-varying features.  I'd refer to the authors to
  1. Temporal FiLM: Capturing Long-Range Sequence Dependencies with Feature-Wise Modulation. (https://proceedings.neurips.cc/paper/2019/file/2afc4dfb14e55c6face649a1d0c1025b-Paper.pdf)
  2. MODELLING BLACK-BOX AUDIO EFFECTS WITH TIME-VARYING FEATURE MODULATION
(https://arxiv.org/pdf/2211.00497)
While I agree that these papers do not apply the temporal FiLM to music generation, the statement as given is nevertheless clearly wrong.

___
- (Line 190) *...that rely on the cross-attention mechanism or element-wise addition, we introduce a more flexible injection mechanism to incorporate melody control into generative models.*

ordering the three methods: cross attention (CA), element-wise addition (EA), and temporal FiLM (TF), according to flexibility, I would come to CA is most flexible, TF is second in the list, EA is last. So this statement looks confusing.

___
- (Line 209) *In addition to external improvements to the condition injection mechanism, we introduce an internal method for improving conditional representations, termed Instance-Adaptive Condition Refinement (IACR).*

For me the proposed IACR is a multiplicative gating which was introduced in the WaveNet paper in 2016. The application context is different, but the principle is the same.

___

- Line 225 and following: *To the best of our knowledge, existing controllable generation methods derive conditional features solely from the conditional input, overlooking their compatibility with the generative model’s hidden state.*

It appears to me that in its generality, the statement is incorrect.  The problem you are trying to solve is not controllable generation but control injection, which is adapting an existing pretrained generator with new controls.

___
- Line 346: *We compare our method... As both methods support only instrumental music generation, we apply them to the same base model, ACE-Step, used in our approach, and ensure consistency in the melody encoder.*

My problem here is that in all tables, you present the methods using only the reference to the original papers as if you have used these methods unchanged. This is misleading. It appears necessary to present this as ACE-Step + SA Controlent, ACE-Step + MuseControLite.

- Line 360: *Our approach achieves the optimal performance across all metrics.* and line 425 * while achieving optimal performance in controlling vocal melodies.*

The term *optimal* implies that the solution has been proven to reach the best possible solution of all possible configurations. I do not see such proof.

**Questions:**

I feel the paper is a significant contribution with interesting results. The main weakness is related to incorrect statements and misleading presentation. Despite these positive findings, in my opinion and in its current form, the paper has to be rejected due to the fact that it sells existing technologies as new contributions.  I suggest the following modifications:

- Correctly cite previous approaches that are equivalent to your *EiLM* and *IACR* methods. These are not new and should be correctly introduced as *Temporal FiLM* or *Time-varying FiLM* and *multiplicative gating*. The applicative context is certainly new, and they can therefore be presented as new contributions. If I did get this wrong and there are fundamental differences that warrant a new term, please explain the differences.

- contextualize the example in the introduction and elaborate on the way your method might be used to achieve what Withney Houston did in terms of reinterpretation. I think that you would basically need to get a new melody contour from a singing performance, whistling, or this could even be a performance with another instrument (a saxophone could be interesting). The model would then have to generate suitable phoneme durations, intensity contours, and voice qualities. I think it would be helpful to discuss the limitations of using synthetic datasets to create such reinterpretations.

In the context of the present article, I feel that the following statement: *We believe that endowing machines with this artistically creative process of reinterpretation represents a promising avenue for empowering music through artificial intelligence.* is unrelated to the proposed method. I would say that if you want to create a model that can reinterpret a song such that it transforms  *a gentle country ballad into a worldwide anthem of deep affection* and if you want the model to learn doing this by itself,  it would be necessary to keep the melody controls incomplete, for example, midi, and rely on the textual descriptions to control how the melody is interpreted. This, however, is not the topic of the present paper.

- Correct or properly contextualize the statement concerning controllable generation and control injection.

- Correct the presentation of the alternative methods in the tables to avoid the misleading impression that you compare to these methods as a whole and not to adaptations of your model that follow the ideas of the references.

- Please reformulate all references to *optimal performance*. Either present a proof that your solution is indeed optimal, or use another term like *the best performance of all methods evaluated*, *superior* compared to the baseline methods.

---

> ### Author Response · Authors · 2025-11-22
> **Response to Reviewer 5LhQ (Part 1/3)**
>
> We are grateful for your review and valuable comments, and we hope our response fully resolves your concerns. Changes are highlighted in the updated manuscript.
> ### Q1. Difference between EiLM and TFiLM
> **[Comparative analysis]**
> While both our EiLM and TFiLM [1,2] are designed for sequential data, they possess a fundamental difference in deriving modulation parameters. TFiLM temporally applies FiLM by partitioning sequences into blocks and using an RNN to generate block-wise modulation parameters recurrently. It is used to capture long-range dependencies from the input itself to self-modulate the network blocks. In contrast, our EiLM generates modulation parameters with the same shape as the conditioning target in a single operation, enabling element-wise modulation. Our frame-wise mapping explicitly avoids temporal interaction when deriving modulation parameters, thereby ensuring strict frame-level correspondence in conditional encoding.
> Therefore, we keep the term 'EiLM' as it more accurately reflects the underlying mechanism, distinguishing our approach from previous methods.
>
> **[Revised version]**
> We cite the relevant TFiLM works and clarify the differences (lines 99–104).
> " Birnbaum et al. (2019) proposed TFiLM, which temporally applies FiLM by partitioning sequences into blocks and using an RNN (Elman, 1990; Graves, 2012) to recurrently generate block-wise modulation parameters. In contrast, we extend FiLM to EiLM (see Figure 1(c)), which generates modulation parameters matching the target dimensions in a single operation without temporal dependency. This design enables element-wise modulation of hidden states, ensuring the temporally aligned injection of melody. "
>
> ### Q2. The Core of IACR and the difference of multiplicative gating
> **[Naming rationale]**
> The core of our Instance-Adaptive Condition Refinement (IACR) module is the idea that the condition vector should be adaptively refined based on the instance-specific hidden states of the generative model, which is the source of its name. We emphasize the necessity of interaction between the condition and the model's internal states, rather than applying a static condition vector [3,4] regardless of the melody information implicitly contained in the hidden states. The specific method to achieve this interaction, specifically the use of the gating mechanism, is secondary to this core concept.
> Since terms like 'multiplicative gating' or 'gated activation unit' [5,6] fail to convey this core idea, we retain the name 'IACR'.
>
> **[Comparative analysis]**
> Although formally similar, our gating mechanism differs from that of WaveNet [5].
> WaveNet's multiplicative gating is applied inside the generative backbone to enable stronger non-linear feature transformation, whereas IACR operates outside the generative backbone to perform instance-adaptive refinement of the melody condition. In WaveNet, the two branches receive inputs from the same source. Conversely, our IACR module takes hidden states and melody conditions as distinct inputs for the two branches to facilitate cross-modal interaction.
>
> **[Revised version]**
> We revise the introduction of IACR, clarifying that the module is implemented based on the gating mechanism from WaveNet, as follows (Line 218-222):
> "...we propose a condition refinement strategy, termed Instance-Adaptive Condition Refinement (IACR), which adaptively refines the condition
> vector based on the hidden states of the generative backbone for improving conditional representations. Our IACR module employs a gating mechanism adapted from WaveNet (van den Oord et al.,2016), where we enable cross-modal interaction between two branches."

---

> > ### Author Response · Authors · 2025-11-22
> > **Response to Reviewer 5LhQ (Part 2/3)**
> >
> > ### Q3. Clarification of Task Definition and Limitations
> >
> > We agree that the Whitney Houston rendition is a complex reinterpretation involving both local (e.g., vibrato, phoneme duration, note transitions) and global adaptations, whereas our work specifically targets the latter.
> >
> > Inspired by image style transfer [7], which reformulates the painting process as texture synthesis while strictly preserving content structure and explicitly ignoring structural deformations in real paintings, we adopt an analogous paradigm for cover song generation. While this remains distant from true artistic covers, it represents a crucial step toward that goal.
> >
> > **[Revised version]**
> > To avoid ambiguity, we explicitly clarify the task in the "Abstract", "Introduction" and "Limitations" sections. The specific revisions are as follows:
> >
> > **[Abstract]** (Line 17-19)
> > "In this work, we reformulate our cover song generation as a conditional generation, which simultaneously generates new vocals and accompaniment conditioned on the original vocal melody and text prompts."
> >
> > **[Introduction]** (Lines 39-49, Task definition and clarification of the gap with real-world examples)
> > "Iconic examples, such as Whitney Houston’s transformative rendition of Dolly Parton’s “I Will Always Love You”, reinterpret the style of the song, evolving a gentle country ballad into a worldwide anthem of deep affection. Given the expressive potential of musical reimagination and cultural significance, we think that cover song generation is a field worthy of exploration.
> > Similar to Whitney Houston’s rendition, musicians creating cover songs may introduce flexible adaptations in local musical elements, such as phoneme durations, vibrato, and note transitions, yielding highly varied and personalized reinterpretations. In this work, we abstract a cover paradigm applicable to arbitrary songs and reformulate our cover song generation as a conditional generation task that performs a global style transfer with text guidance while preserving the source vocal melody contour and disregarding local customized adaptations."
> >
> > **[Limitations]** (Line 498-509, Task scope, limitations of using synthetic data, and future directions)
> > "In this work, we exclude the song-specific, local adaptations (e.g., variations in phoneme durations, vibrato, and note transitions) that musicians may introduce when creating covers. ... Additionally, AI-generated songs lack the expressive subtlety of human singing and fine-grained vocal technique annotations, preventing our model from achieving the micro-level expressiveness typical of professional covers.
> > Future research could incorporate such fine-grained control by developing song generation models capable of understanding time-aligned musical prompts for precise adaptation control. More ideally, by constructing real paired original-cover datasets, models could learn to autonomously reinterpret an incomplete melody, employing both global and local adaptations to convey distinct emotions and styles."
> >
> > ### Q4. Path Toward Whitney Houston-style Covers
> > We obtain modified melody contours by applying linear interpolation (for tempo adjustment) and value scaling (for key transposition) to the original F0 sequence.
> > Our model successfully rendered the song adhering to the new contours, as demonstrated in Section 5 of the [updated demo](https://vvanonymousvv.github.io/SongEcho_updated/). While our model does not autonomously generate the complex local adaptations of Whitney Houston's rendition, these results demonstrate its potential to achieve such expressive reinterpretation when coupled with melody editing tools or human performance.
> > We add relevant discussion in the "Discussions and Limitations" section (Line 501-503):
> > "One promising avenue toward Whitney Houston-style expressive covers is to combine our method with melody editing tools or human creators. Local creative modifications can be introduced via external editing or live performance, after which our model generates a globally reinterpretation of the revised melody contour."
> >
> > ### Q5. Clarification of the Model's Reinterpretation Capability
> > We remove the misleading statement (originally Line 42-43): "We believe that endowing machines with this artistically creative process of reinterpretation represents a promising avenue for empowering music through artificial intelligence".  In this work, the reinterpretation capability that we aim to endow the model with lies in its ability to autonomously determine how to integrate a global, descriptive text with the vocal melody, generating a stylistically cohesive song that adheres to both constraints.

---

> > > ### Author Response · Authors · 2025-11-22
> > > **Response to Reviewer 5LhQ (Part 3/3)**
> > >
> > > ### Q6. Flexibility Comparison of Injection Mechanisms
> > > We agree with the reviewer's ranking regarding the flexibility of the three mechanisms. A detailed comparison of them is provided in the Introduction (Lines 84-90).
> > > We revise the description of EiLM (originally Line 190) as follows (Line 197-199):
> > > "Unlike prior temporally controllable music generation methods (Wu et al., 2024; Tsai et al., 2025; Hou et al., 2025; Yang et al., 2025), which rely on cross-attention or element-wise addition, we explore the application of FiLM (Perez et al., 2018) for melody injection and extend it to EiLM."
> > >
> > > ### Q7. Other Corrected Expressions
> > > We make the following corrections and revisions:
> > > (1) We replace "controllable generation" with "control injection".
> > > (2) We now refer to the baselines by their full names: 'ACE-Step + SA ControlNet' and 'ACE-Step + MuseControlLite'.
> > > (3) We remove all instances of "optimal performance" and utilize terms like "superior results compared with the baselines".
> > > (4) We check the manuscript and revise other ambiguous or unclear expressions (highlighted in the updated version).
> > >
> > > Thank you again for the constructive feedback. We are happy to address any further questions or concerns you may have.
> > >
> > > [1] Temporal FiLM: Capturing Long-Range Sequence Dependencies with Feature-Wise Modulations. NeurIPS 2019
> > > [2] Modelling Black-Box Audio Effects with Time-Varying Feature Modulation. ICASSP 2023
> > > [3] MuseControlLite: Multifunctional Music Generation with Lightweight Conditioners. ICML 2025
> > > [4] Editing music with melody and text: Using controlnet for diffusion transformer. ICASSP 2025
> > > [5] WaveNet: A Generative Model for Raw Audio. SSW 2016
> > > [6] Conditional Image Generation with PixelCNN Decoders. NIPS 2016
> > > [7] Arbitrary Style Transfer in Real-Time with Adaptive Instance Normalization. ICCV 2017

---

### Official Review · Reviewer_ACB8 · 2025-10-30

**Soundness:** 3
**Presentation:** 3
**Contribution:** 2
**Rating:** 6
**Confidence:** 4

**Summary:**

The paper employs an Instance-Adaptive Element-wise Linear Modulation (IA-EiLM) framework to enable controllable song generation. Specifically, Element-wise Linear Modulation (EiLM) improves the temporal alignment of melody control, while Instance-Adaptive Condition Refinement (IACR) enhances the condition representation. To address the scarcity of large-scale, open-source full-song datasets, the authors construct a large-scale dataset.

**Strengths:**

This paper proposes Instance-Adaptive Element-wise Linear Modulation (IA-EiLM), which comprises the EiLM and Instance-Adaptive Condition Refinement (IACR), enhancing the condition injection mechanism and conditional representation, respectively.
This paper introduces Suno70k, an open-source AI song dataset enriched with detailed annotations, including enhanced tags and lyrics.

**Weaknesses:**

One of the paper's claimed innovations, EiLM, appears to be a relatively trivial extension of FiLM. This leaves IACR as the paper's primary technical insight, which may render the overall technical novelty somewhat limited.

**Questions:**

Can the technique proposed in this paper be extended to autoregressive singing voice synthesis?

---

> ### Author Response · Authors · 2025-11-22
> **Response to Reviewer ACB8**
>
> We are grateful for your review and valuable comments, and we hope our response fully resolves your concerns. Changes are highlighted in the updated manuscript.
> ### Q1. Novelty and necessity of EiLM
> **[Necessity]** Technically, while EiLM may be straightforward to implement, it represents an elegant and necessary extension of FiLM. This adaptation is essential because the original FiLM is limited to global conditioning, whereas melody control in music generation requires time-varying conditioning.
>
> **[Novel application]** To the best of our knowledge, EiLM marks the first application of FiLM-style modulation to temporally controllable music generation. Furthermore, in contrast to TFiLM [1], which applies FiLM recurrently via an RNN to handle sequential features, EiLM avoids such recurrent computation, which would be unnecessarily complex and computationally expensive for our task.
>
> **[Effectiveness]** Ablation studies further demonstrate that our EiLM effectively enhances melody controllability.
>
> ### Q2. Generalizability to Autoregressive generation models
>
> **[Theoretical analysis]** Our method can be adapted to autoregressive (AR) models, as both architectures rely on token-based representations compatible with our token-level injection mechanism. The primary adaptation for AR involves enforcing causal constraints, injecting melody conditions token-by-token during the autoregressive generation process.
>
> **[Supporting work]** This feasibility is supported by ControlAR [2], which demonstrates that injecting spatial conditions token-by-token for autoregressive generation models effectively achieves spatially controllable image generation. We believe this is a valuable extension that deserves further exploration in future work.
>
> Thank you again for the constructive feedback. We are happy to address any further questions or concerns you may have.
>
> [1] Temporal FiLM: Capturing Long-Range Sequence Dependencies with Feature-Wise Modulations. NeurIPS 2019
> [2] ControlAR: Controllable Image Generation with Autoregressive Models. ICLR 2025

---

### Official Review · Reviewer_PTPX · 2025-10-31

**Soundness:** 3
**Presentation:** 3
**Contribution:** 3
**Rating:** 8
**Confidence:** 4

**Summary:**

This paper investigates the interesting task of cover song generation and proposes a parameter-efficient framework based on ACE-Step to enable melody-controlled song generation. It introduces EiLM, an efficient condition injection mechanism, and IACR, which dynamically adapts conditioning to the model’s hidden states. The authors also release the Suno-70k dataset. The paper is clearly written, and the experiments are thorough and convincing.

**Strengths:**

1. The paper formalizes an intersting task—melody-controlled cover song generation—and carefully discusses the limitations of existing condition-injection mechanisms in NAR frameworks. The proposed IA-EiLM achieves superior performance in parameter efficiency, precise temporal control, and melody adherence.
2. The experiments and ablation studies are comprehensive and well-designed.
3. The Suno-70k dataset represents a meaningful contribution to the research community.
4. The presentation is clear, with well-designed figures and an easy-to-follow narrative.

**Weaknesses:**

1. The paper lacks details on the exact form of melody input. If it only uses pitch sequences, how is the alignment between notes and lyrics ensured? Why was this particular representation chosen, and were other melody representations considered?
2.  How does the model handle conflicts between text tags and melody? Since a melody implicitly encodes stylistic attributes, it would be useful to clarify how such inconsistencies are resolved.
3.  Although SongEval Aesthetics Metrics are included in the appendix, it remains unclear whether adding melody control compromises ACE-Step’s original generation quality. A demo comparison with ACE-Step would be helpful.
4.  Minor: In the section “Why is IACR necessary?”, the symbols E, T, and M are undefined, which makes the discussion slightly confusing.

**Questions:**

See Weakness section above.

---

> ### Author Response · Authors · 2025-11-22
> **Response to Reviewer PTPX**
>
> We are grateful for your review and valuable comments, and we hope our response fully resolves your concerns. Changes are highlighted in the updated manuscript.
>
> ### Q1.Details of melody input & Note-lyrics alignment
>
> **[Details of melody input]**
> We condition on the RVMPE-extracted F0 sequence, which is preprocessed by normalizing only its voiced components (50-900Hz) and concatenating the result with a derived binary voiced/unvoiced flag ($flag_{uv}$) to form the final melody feature.
>
> **[Note-lyrics alignment]**
> Our method achieves lyric-to-note alignment without explicit duration modeling or external aligners. The $flag_{uv}$ accurately delineates voiced regions, and visualization (see Fig.3 in Appendix B.1) shows that phoneme transitions consistently align with inflection points in the F0 curve. By jointly optimizing melody (F0) and linguistic content (phonemes) during source song reconstruction, the model leverages the strong phoneme-note dependencies captured by its pretrained backbone to implicitly construct a phoneme layout along the F0 timeline.
>
> **[Melody representations selection]**
> F0 sequence is widely used in Sing Voice Conversion [1,2] for its accurate preservation of melodic details. In contrast, features like discrete notes, which lack explicit transition information, may lead to alignment issues and compromise the naturalness of the generated songs. Therefore, our method leverages the F0 sequence to achieve note-lyric alignment and high-fidelity song generation.
>
> ### Q2. Performance When Tags Conflict with the Melody
> We found that the model prioritizes the source melody when the provided style tags conflict with the melody condition. As shown in Section 3 of the [updated demo](https://vvanonymousvv.github.io/SongEcho_updated/), when transferring a Rap track to a "Soul folk acapella" style, the model only successfully alters the accompaniment style. The resulting song retains its Rap identity because the original melody is preserved.
> In practice, musicians typically respect the inherent constraints of the original melody, generally avoiding target styles that create major clashes with the source song.
>
> ### Q3. Results of ACE-Step
>  We add the generated results of the original model to the [updated demo](https://vvanonymousvv.github.io/SongEcho_updated/) (Section 1) for reference. As shown in the demo, integrating melody control into Ace-Step via our method imposes negligible degradation on the original generation quality. Furthermore, driven by high-quality melody guidance, the model exhibits superior aesthetic performance.
>
> ### Q4. Clarify undefined symbols (Line 241, 247)
> To avoid ambiguity in the main text, we have renamed the conditional mapping function from $T$ to $F$. Furthermore, we define $E_m$ as a hypothetical melody encoder designed to extract melody information from the hidden states.
>
> Thank you again for the constructive feedback. We are happy to address any further questions or concerns you may have.
>
> [1]FreeSVC: Towards Zero-shot Multilingual Singing Voice Conversion. ICASSP 2025
> [2]Everyone-Can-Sing: Zero-Shot Singing Voice Synthesis and Conversion with Speech Reference. ICASSP 2025

---

### Author Response · Authors · 2025-12-02
**Summary of Revisions**

We sincerely thank all four reviewers for their valuable comments, which have significantly strengthened our work. Below, we summarize the major revisions (all changes are highlighted in the revised manuscript). Detailed responses are provided in the individual reviewer threads.

**1. Clarification of Task Definition and Limitations**
* **Addresses:** 5LhQ (W1 (Weakness 1)), dAnD (W4; Q2 (Question 2)).
* We explicitly clarify the task in Sec. 1 and Sec. 5.5.
**[Introduction]** (Line 44-48)
"Similar to Whitney Houston's rendition, musicians creating cover songs may introduce flexible adaptations in local musical elements, such as phoneme durations, vibrato, and note transitions, yielding highly varied and personalized reinterpretations. In this work, we abstract a cover paradigm applicable to arbitrary songs and reformulate our cover song generation as a conditional generation task that performs a global style transfer with text guidance while preserving the source vocal melody contour and excluding local customized adaptations."
**[Limitations]** (Line 498-499)
"In this work, we exclude the song-specific, localized modifications (e.g., variations in phoneme durations, vibrato, and note transitions) that musicians may introduce when creating covers. ..."

**2. Citations and Clarifications of Related Work**
* **Addresses**: 5LhQ (W2,4; Q1)
* We cite the relevant TFiLM works and clarify the differences (lines 99–104).
"Birnbaum et al. (2019) proposed TFiLM, which temporally applies FiLM by partitioning sequences into blocks and using an RNN to recurrently generate block-wise modulation parameters. In contrast, we extend FiLM to EiLM (see Figure 1(c)), which generates modulation parameters matching the target dimensions in a single operation without temporal dependency..."
* We cite WaveNet and clarify that IACR is our proposed conditioning strategy with a distinct motivation (Line 218-222):
"...we propose a condition refinement strategy, termed Instance-Adaptive Condition Refinement (IACR), which adaptively refines the condition vector based on the hidden states of the generative backbone for improving conditional representations. Our IACR module employs a gating mechanism adapted from WaveNet (van den Oord et al.,2016), where we enable cross-modal interaction between two branches."

**3. Baseline Details & Additional Comparisons**
* **Addresses**: dAnD (W1–3; Q1)
* Add baseline details in Appendix C.1.
* Add inference results for ACE-Step + MuseControlLite using Multiple Classifier-Free Guidance (See Table 6).
* Perform full-parameter training on the control branch of ACE-Step + SA ControlNet using BF16 mixed precision and add results in Tables 1–3.

**4. Validity of Equation 6 Regarding Audio Copying**
* **Addresses**: dAnD (Round 2, Q3)
* We strengthen the theoretical analysis, which further shows that static conditioning makes Eq. 6 underconstrained for our task (Line 251, Appendix B.2).

**5. Path Toward Whitney Houston-style Covers & Limitations of Synthetic Datasets**
* **Addresses**: 5LhQ (Q2)
* We perform global editing on the melody contours. As shown in Sec. 5 of the [updated demo](https://vvanonymousvv.github.io/SongEcho_updated/), our model successfully renders the song while adhering to the new contours.
* Based on the above, we add relevant discussion in Sec. 5.5 (Lines 501-503): "One promising avenue toward Whitney Houston-style expressive covers is to combine our method with melody editing tools or human creators. Local creative modifications can be introduced via external editing or live performance, after which our model generates a global reinterpretation of the revised melody contour."
* We also clarify the limitations of using synthetic data in Sec. 5.5 (Line503-505): "Additionally, AI-generated songs lack the expressive subtlety of human singing and fine-grained vocal technique annotations, preventing our model from achieving the micro-level expressiveness typical of professional covers."

**6. Enhanced Demo**
* **Addresses**: PTPX (W2-3), dAnD (Q3)
* The [updated demo](https://vvanonymousvv.github.io/SongEcho_updated/) now includes the original ACE-Step results, the Results of Tag-Melody Conflict, and the Inpainting & Outpainting results.

**7. Details of Melody Input & Note-lyrics alignment**
* **Addresses**: PTPX (W1)
* In Appendix B.1, we provide a detailed description of the melody input and analyze the note-lyrics alignment mechanism with a visualization of f0 contour-lyrics alignment.

**8. Other Presentation Improvements**
* **Addresses**: 5LhQ (W3,5,6,7; Q3,4,5)
* Remove the misleading statement about reinterpretation and EiLM flexibility.
* Replace "controllable generation" with "control injection" (Line 234).
* Refer to the baselines by their full names throughout the revised manuscript: ACE-Step + SA ControlNet and ACE-Step + MuseControlLite.
* Replace all instances of "optimal performance" with terms like "superior results compared to baselines" (Line 381, 417, 436).

---

### Author Response · Authors · 2025-12-02
**Summary for the Area Chair**

Dear Area Chair,

Below, we summarize the positive feedback and the resolution of key concerns (with reply from Reviewer dAnD).

**1. Summary of Positive Feedback**

**[Contribution: Significant and Interesting]**
PTPX: `investigates the interesting task`
5LhQ: `I feel the paper is a significant contribution with interesting results.`
dAnD: `This is an interesting paper.`

**[Methodology: Superior and Efficient]**
PTPX: `parameter-efficient framework` `achieves superior performance...`
dAnD: `using fewer parameters..., it demonstrates better performance...`.

**[Dataset: Large-Scale and Detailed Annotations]**
All reviewers affirm the contribution of our suno70k dataset, emphasizing it as `large-scale` (ACB8, 5LhQ), `detailed annotations` (ACB8, 5LhQ, dAnD), and `a meaningful contribution` (PTPX).

**[Experiments: Comprehensive and Effective]**
PTPX:`The experiments and ablation studies are comprehensive and well-designed.`
5LhQ: `successful evaluation of the proposed methods`.

**[Analysis: In-Depth and Critical]**
PTPX: `carefully discusses the limitations of existing condition-injection mechanisms`
dAnD: `Provides a critical analysis across different conditioning mechanisms`.

**2. Score Increase from Reviewer dAnD (4→6)**
After we addressed the initial concerns, the reviewer raised new valuable questions.
Through two rounds of rigorous discussion, Reviewer dAnD concluded: "Nice catch! The authors' reply fully answered my questions, I raised my score".

**3. Concerns of Reviewer 5LhQ and Our Responses**
Reviewer 5LhQ explicitly states: `I feel the paper is a significant contribution with interesting results. The main weakness is related to incorrect statements and misleading presentation.` Coupled with the praised `successful evaluation of the proposed methods,` this confirms that the reviewer validates our contribution and experiments, limiting concerns to presentation and clarification.
We carefully revise the presentation point-by-point according to the reviewer's suggestions in the revised manuscript and summarize the key points here.

**[Clarification on Our Method]**
The reviewer comments on our method as follows: `The applicative context is certainly new, and they can therefore be presented as new contributions.` However, we respectfully clarify that our method involves distinct contributions beyond existing methods. For the related works raised by the reviewer, we cite them and clearly explain the differences in the revised manuscript. We clarify the fundamental differences as follows:

* EiLM vs. TFiLM [1]:
The two differ significantly in their parameter generation mechanism. TFiLM temporally applies FiLM via RNNs on sequence blocks to recurrently generate block-wise modulation parameters. In contrast, our EiLM applies a single frame-wise operation to generate element-wise modulation parameters matching the target's exact shape without temporal interaction.

* IACR vs. Multiplicative Gating [2]:
`IACR (Instance-Adaptive Condition Refinement)` explicitly targets the underconstrained nature of static conditioning in our task by dynamically interacting between the condition and the hidden states.
`Multiplicative gating` cannot express this core idea and is the basic component we adopt in this work to implement cross-modal interaction. Theoretically, cross-attention can also achieve this interaction, but it introduces unnecessary computation since our task requires only frame-independent interaction.
***Note: We conduct a rigorous discussion with Reviewer dAnD proving the theoretical necessity of IACR as a dynamic conditioning strategy (See Round 2, Q3). This further shows that the underlying principle of IACR is independent of multiplicative gating.***

**[Refinement of Task Definition]**
Regarding the Whitney Houston example, our task focuses on global style adaptation while excluding local customized adaptations(e.g., vibrato, phoneme duration). In the revised manuscript, we clarify the task scope in the Introduction and explicitly outline the method’s applicability in the Discussion and Limitations section. In the Introduction section, we state: `...reformulate our cover song generation as a conditional generation task that performs a global style transfer with text guidance while preserving the source vocal melody contour and excluding local customized adaptations.`
***Note: Reviewer dAnD raised a similar concern regarding task scope and was satisfied with our revised definition.***


**4. Response to Reviewers PTPX and ACB8**
We revised the manuscript following Reviewer PTPX's suggestions and answered all questions from Reviewers PTPX and ACB8 in the individual responses.

We sincerely thank you for your time and effort on our paper.

References
[1] Temporal FiLM: Capturing Long-Range Sequence Dependencies with Feature-Wise Modulations. NeurIPS 2019
[2] WaveNet: A Generative Model for Raw Audio. SSW 2016

Best Regards,

Submission 4120 Authors

---

### Meta-Review · Area_Chair_PX82 · 2025-12-22

**Summary:**

In the initial round of reviews, two reviewers evaluated the paper as above the acceptance threshold (scores 8,6) and two evaluated it below the threshold (score 4,2). The main concerns were: (1) limited technical novelty (ACB8, 5LhQ); (2) inappropriate framing of the task being solved as “cover song generation” (dAnD, 5LhQ); (3) lack of details on melody injection mechanism and its effect on the model’s original quality (PTPX); (4) issues with the implementation of the baseline methods.

After the rebuttal, reviewer dAnD indicated they would like to raise the score (from 4 to likely 6). No response was recorded from the other reviewers.

The AC agrees with the reviewers’ criticism on the limited technical novelty and on the framing of the task as “cover song generation” despite addressing a narrower task in practice. Nevertheless, regardless of how one chooses to call this task, the reviewers seem to agree that its treatment is novel. Based on this, on the overall positive evaluations, and on the clarifications and disclaimers added to the manuscript, the AC sees the paper as appropriate for publication in this ICLR.

**Reviewer Concerns:**

Most major concerns have been properly addressed and misunderstandings have been clarified by the rebuttal. Regarding the point about the limited technical novelty of EiLM with respect to FiLM, the authors acknowledged that the novelty is not technical but explained it is rather conceptual (FiLM introduces global modulation parameters and EiLM introduces time-varying parameters). Regarding the framing of the method, despite the clarifications added to the manuscript, the AC still views the manuscript as slightly overselling. For example, the term “cover song…” appears in the title. The authors are encouraged to consider changing this to “towards cover song generation…”.

**Reviewer Scores:**

**PTPX : score 8.**

This is also the original score. The rebuttal seems to have addressed all the reviewer’s concerns.

**ACB8: score 6.**

This is also the original score. The AC predicts the reviewer would have not changed their minds regarding the point they raised about technical novelty.

**5LhQ: score 4.**

The reviewer’s original score was 2, mainly because of concerns about limited technical novelty and about the framing of the task. Nevertheless, the reviewer indicated that the paper is a significant contribution with interesting results. The authors added clarifications and disclaimers to the manuscript, which the AC believes would at least partially satisfy the reviewer, hence the estimated raise of score from 2 to 4.

**dAnD : score 6.**

The initial score was 4 but following the rebuttal, the reviewer stated they would raise the score (likely to 6).

---

### Decision · Program_Chairs · 2026-01-26

Accept (Poster)